# Identifying and validating ITGB2 and HNRNPAB as diagnostic biomarkers in chronic obstructive pulmonary disease using bioinformatics and Integrated Machine Learning Methods

Fengjun Zhang[1☯], Hui Li[2☯], Fan Wu[1], Dexian Xian[3], Feng Chen[4], Wenchang Xu[3], Yuchen He[5], Xiaodan Liu[2‡]*, Wei Zhang[6,7,8‡]*

**1** Department of Pulmonary Diseases, Shuguang Hospital Affiliated to Shanghai University of Traditional Chinese Medicine, Shanghai, China, **2** School of Rehabilitation Science, Shanghai University of Traditional Chinese Medicine, Shanghai, China, **3** The First Clinical Medical, Shandong University of Chinese Medicine, Jinan, China, **4** Department of Integrated Chinese and Western Medicine, Yantai Yuhuangding Hospital Affiliated to Qingdao University, Yantai, China, **5** Department of rehabilitation, Shuguang Hospital Affiliated to Shanghai University of Traditional Chinese Medicine, Shanghai, China, **6** Department of Pulmonary Diseases, Shuguang Hospital, Shanghai Institute of Infectious Diseases and Biosecurity, Shanghai University of Traditional Chinese Medicine, Shanghai, China, **7** Zhang Wei Baoshan famous traditional Chinese medicine inheritance studio, Shanghai Baoshan Hospital of Integrated Chinese and Western Medicine, Shanghai, China, **8** Institute of Infectious Diseases, Shanghai Institute of Traditional Chinese Medicine, Shanghai, China

☯ These authors contributed equally to this work.
‡ WZ and XL also contributed equally to this work.
* zhangw1190@sina.com (WZ); hzhp403@126.com (XL)

## Abstract

### Background and Aim

COPD is a common respiratory disease characterized by progressive airflow restriction that severely affects patients' quality of life and leads to significant mortality rates worldwide. This study aims to strengthen the early diagnosis of COPD and develop personalized treatment strategies.

### Methods

The methodology involved a comprehensive approach, including differential gene expression analysis, weighted gene co-expression network analysis (WGCNA), functional enrichment analysis, and machine learning techniques. Data from the combined datasets GSE37768 and GSE38974 were utilized to identify differentially expressed genes (DEGs). The machine learning integrated model was employed to screen for diagnostic molecular biomarkers related to COPD. Additionally, pathway analysis, transcription factor gene regulatory network analysis, immune cell composition analysis using CIBERSORT, and mendelian randomization analysis were conducted to elucidate the molecular mechanisms and potential biomarkers for COPD.

**Data availability statement:** The data that support the findings of this study are available in Gene Expression Omnibus at https://www.ncbi.nlm.nih.gov/geo/, reference number GSE37768, GSE38974, GSE212331, GSE148004, GSE1650. These data were derived from the following resources available in the public domain: - GSE37768, https://www.ncbi.nlm.nih.gov/geo/query/acc.cgi?acc=GSE37768 - GSE38974, https://www.ncbi.nlm.nih.gov/geo/query/acc.cgi?acc=GSE38974 - GSE212331, https://www.ncbi.nlm.nih.gov/geo/query/acc.cgi?acc=GSE212331 - GSE148004, https://www.ncbi.nlm.nih.gov/geo/query/acc.cgi?acc=GSE148004 - GSE1650, https://www.ncbi.nlm.nih.gov/geo/query/acc.cgi?acc=GSE1650.

**Funding:** Project supported by Shanghai Municipal Science and Technology Major Project (ZD2021CY001), Shanghai Key Laboratory of Internal Medicine of Traditional Chinese Medicine (grant numbers 20DZ2272200), Zhang Wei's Inheritance and Innovation Studio of Traditional Chinese Medicine(2025CXGZS-01), Zhang Wei Baoshan famous traditional Chinese medicine inheritance studio (BSMZYGZS-2024-01) and Zhang Wei Medical Technology Doctor Site Construction-Respiratory therapy Technology Direction (grant numbers A1-N23-204-0405).

**Competing interests:** The authors have declared that no competing interests exist.

Finally, we validated the model using Polymerase Chain Reaction (PCR), immunohistochemistry (IHC) and Immunofluorescence (IF).

## Results

This study employed bioinformatics and Integrated Machine Learning Methods to identify ITGB2 and HNRNPAB as potential related targets for COPD. Subsequent verification through PCR, IHC, and IF experiments confirmed that ITGB2 and HNRNPAB were key biomarkers for COPD. Pathway analysis revealed that ITGB2 and HNRNPAB were mainly involved in immune responses and metabolic pathways.

## Conclusion

This comprehensive study presents an in-depth investigation of the molecular mechanisms of COPD and identifies candidate exploratory biomarkers for further research toward early diagnosis and potential personalized treatment strategies. In future studies, the identified exploratory biomarkers should be validated in larger cohorts and their therapeutic significance explored.

## Introduction

Chronic Obstructive Pulmonary Disease (COPD) is a progressive respiratory condition marked by ongoing airflow restriction, mainly due to long-term exposure to harmful substances or gases, particularly from cigarette smoke [1]. This illness significantly impacts both patients and society, leading to high rates of illness, death, and economic burdens related to healthcare costs and lost productivity [2]. Current treatments for COPD mainly consist of bronchodilators, corticosteroids, and pulmonary rehabilitation [3,4]. However, these approaches often only alleviate symptoms rather than stop the disease from worsening. Additionally, the effectiveness of existing therapies can vary among patients, and they may come with side effects while failing to tackle the root causes of the disease. Therefore, there is an urgent need for new treatment strategies and a better understanding of the molecular and genetic factors involved in COPD, highlighting the importance of this research.

There is increasing evidence that systemic immune response is deeply involved in the pathological changes and disease progression of COPD [5–7]. Previous research has demonstrated that both the enhancement and suppression of immune responses are intricately linked to the pathophysiology of COPD. In patients with mild-to-moderate stable COPD, there is a corresponding increase in tissue-resident immune cells [8]. The progressive worsening of COPD is associated with suppressed immune responses, characterized by impaired immune cell function and reduced cell numbers [9,10]. Notably, immune infiltration biomarkers have emerged as promising prognostic indicators for COPD, offering insights into disease severity and progression [11,12]. Furthermore, these biomarkers hold potential as prospective therapeutic targets, highlighting the significance of immune modulation in the management of COPD. The

exploration of these associations underscores the innovative approach of this research, aiming to elucidate the dual role of immune responses in COPD and to pave the way for novel therapeutic strategies.

Notably, the integration of machine learning techniques with Mendelian randomization offers a novel approach to mitigate confounding factors and establish causal inferences, thereby strengthening the validity of identified biomarkers [13]. The potential implications of this research are significant, as it not only advances our understanding of disease mechanisms but also paves the way for the development of targeted therapeutic strategies and personalized medicine.

This study aims to identify key molecules and pathway mechanisms associated with COPD using integrative bioinformatics approaches, while also developing a predictive model for early diagnosis. By incorporating differential expression analysis, weighted gene co-expression network analysis (WGCNA), machine learning frameworks, immune infiltration profiling, transcription factor regulatory network construction, enrichment analyses, Mendelian randomization, gene set variation analysis (GSVA) and Polymerase Chain Reaction (PCR), we seek to uncover the biological processes and pathways driving COPD progression. Ultimately, our goal is to provide a preliminary theoretical foundation for future exploration of targeted interventions and potential personalized therapeutic strategies. Fig 1 showed the flow chart of this study.

## 2. Methods and materials

### 2.1. COPD datasets acquisition

Relevant gene expression profiles were identified using the keyword "COPD" in the GEO database. The array datasets GSE37768, GSE38974, GSE212331, GSE148004, and GSE1650 were downloaded for analysis. Specifically, GSE37768 includes 18 COPD lung tissue samples and 9 non-smoking lung tissue samples (with samples from smoking patients excluded), while GSE38974 comprises 23 COPD samples and 9 healthy control samples (platform: Agilent-014850 Whole Human Genome Microarray 4x44K G4112F) [14]. Meanwhile, The GSE212331 dataset contains 72 COPD patient samples and 15 healthy control samples [15], GSE148004 includes 7 COPD patient samples and 10 healthy control samples

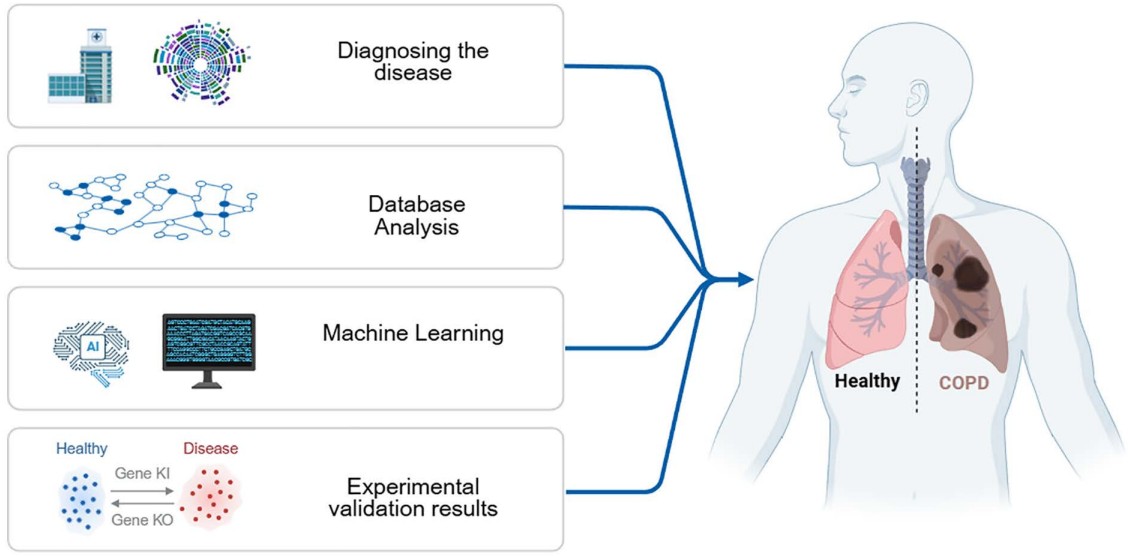

**Fig 1. Workflow diagram of the current study.**

[16], and GSE1650 consists of 18 COPD patient samples and 12 healthy control samples [17]. These three datasets were used as independent cohorts to evaluate the performance of 113 machine learning models.

## 2.2. Data preprocessing and merging

The raw gene expression matrix from the The Gene Expression Omnibus (GEO) dataset was preprocessed and sorted using Perl to ensure accurate mRNA data. The R package "limma" (version 3.54.2) was employed to normalize and process the mRNA expression matrix [18,19]. Additionally, the "SVA" package was utilized to integrate the GSE37768 and GSE38974 datasets and eliminate batch effects, further enhancing data consistency.

## 2.3. Identification of DEGs in COPD

The R package "limma" was utilized to identify DEGs from the merged GSE37768 and GSE38974 datasets. For the microarray expression data, genes were classified as DEGs if they met the criteria of |log2 FoldChange| ≥ 0.585 and an adjusted p-value < 0.05 [20]. Visualization of DEGs, including volcano plots and heatmaps, was generated using the R packages "ggplot2" and "pheatmap," respectively.

## 2.4. Weighted gene co-expression network analysis (WGCNA)

The WGCNA algorithm was employed to categorize genes and assess the relationship between gene modules and clinical traits [21]. WGCNA was conducted on the variable genes from the merged GSE37768 and GSE38974 datasets. Module connectivity was determined based on gene correlation within each module, allowing for the exploration of co-expression similarity. To investigate the association between gene expression modules and clinical traits, correlation coefficients and p-values were calculated, with modules considered significant if the p-value was less than 0.05. A heatmap was used to visualize these relationships.

Additionally, the "venn" package in R was used to generate Venn diagrams comparing DEGs with co-expression module genes. Genes shared between these sets were identified as potential key contributors to COPD.

## 2.5. Functional enrichment analysis

Functional enrichment analysis encompassed both GO [22,23]and KEGG [24] pathway analyses. GO analysis was conducted across three domains: biological processes, molecular functions, and cellular components, while KEGG pathways were used to interpret metabolic pathways and assess gene and genomic functions. Both approaches play a crucial role in understanding gene function during enrichment analysis.

The R software package "org.hs.egg.db" (version 3.16.0) was used as a reference background for the GO and KEGG pathway enrichment analysis process. In addition, we used the R package "clusterProfiler" (version 4.6.2) to obtain functional enrichment results. If the GO term and the KEGG pathway meet the p < 0.05 condition, this result is considered significantly enriched.

## 2.6. Integrated machine learning framework to identify and construct characteristic genes and diagnostic models of COPD

Using an integrated machine learning approach, we identified and selected pivotal genes associated with COPD to construct a molecular diagnostic model that differentiates between control subjects and COPD cases. The development of the COPD diagnostic model was based on a training cohort derived from the merged datasets GSE37768 and GSE38794, employing a framework that integrates twelve machine learning algorithms: Least Absolute Shrinkage and Selection Operator (LASSO), ridge regression, elastic network (Enet), support vector machine (SVM), gradient boosting with component-wise linear models (glmBoost), stepwise generalized linear model (Stepglm), linear discriminant analysis

(LDA), random forest (RF), gradient boosting machine (GBM), partial least squares regression for generalized linear models (plsRglm), eXtreme gradient boosting (XGBoost), and Naive Bayes [13,25]. Detailed descriptions of these algorithms are summarized in the S1 File.

Within this integrated computational framework, four feature selection algorithms—Lasso, RF, Stepglm, and glmBoost—were employed to benchmark the model, while the remaining algorithms were utilized for model fitting, resulting in a total of 113 machine learning combinations.

To avoid overfitting and data leakage, strict separation between training and external test cohorts was implemented. The integrated machine learning framework was constructed exclusively on the merged training cohort (GSE37768 and GSE38974). 10-fold cross-validation (10-CV) was applied during model training and hyperparameter optimization, with feature selection procedures (Lasso, RF, Stepglm, glmBoost) nested within each cross-validation fold. All preprocessing steps including batch correction and normalization were conducted only within the training subset of each fold. Three completely independent datasets (GSE212331, GSE148004, and GSE1650) were used as external test sets and were not involved in gene screening, feature selection, model construction, or parameter tuning.

The R package "pROC" was employed to evaluate the predictive performance of the diagnostic model and calculate the area under the receiver operating characteristic curve (AUC). The machine learning combination achieving the highest average AUC in both the training and testing cohorts was designated as the optimal model, referred to as MS. The experimental setup of the integrated machine learning framework, including the use of R packages, cross-validation, and hyperparameter optimization, is detailed in the supplementary materials.

### 2.7. Transcription Factor (TF)-gene regulatory network construction

The JASPAR database (http://jaspar.genereg.net/) [26], accessed via the NetworkAnalyst 3.0 platform [27], was utilized to construct a co-regulatory network of transcription factors (TFs) associated with COPD diagnostic biomarkers. Visualization of this network was performed using Cytoscape software. Based on twelve identified COPD diagnostic biomarkers, we identified transcription factors from the JASPAR database that regulate COPD-related pathways and gene expression levels, leading to the development of a TF gene regulatory network.

### 2.8. Immune infiltration analysis

We employed the CIBERSORT algorithm to identify the immune cell composition within the COPD gene expression matrix. Using the CIBERSORT software, we calculated the proportions of 22 immune cell types between the COPD and control groups, visualizing the differential expression of immune cells through bar and box plots. The sum of the percentages of the 22 immune cell types in both groups was constrained to 1, with 1,000 simulations conducted and significance set at $p < 0.05$ [28].

### 2.9. Mendelian randomization (MR) analysis

We conducted Mendelian Randomization (MR) analysis to identify pivotal genes for COPD, assessing their potential as pathogenic and therapeutic candidates to enhance the model's therapeutic efficacy [13]. The instrumental variables (SNPs) for ITGB2 and HNRNPAB protein expression were derived from large-scale human plasma proteome quantitative trait locus (pQTL) data from the UK Biobank and deCODE genetics proteome database. The R package "TwoSampleMR" [29] facilitated a two-sample MR analysis to investigate the causal relationship between gene expression (exposure) and COPD risk (outcome). We retrieved the deCODE database (https://www.decode.com/summarydata/) protein quantitative trait loci (pQTL) as exposure data, this study selected FinnGen alliance of COPD outcomes data (R10 release, on October 13, 2024) [30]. The analysis focused on the "COPD" phenotype, which comprised 166,401 Finnish adult participants, including 20,066 cases and 338,303 controls. Adjustments were made for sex, age, the first ten principal components, and genotyping batches.

In accordance with MR assumptions [31], we selected single nucleotide polymorphisms (SNPs) significantly associated with protein expression, applying a threshold of $p < 5 \times 10^{-8}$. These SNPs served as instrumental variables (IVs) for the two-sample MR analysis. To mitigate linkage disequilibrium (LD), we excluded SNPs with an LD-$R^2$ greater than 0.01 within a 10,000 Kb window. The exposure and outcome data were subsequently harmonized. Five methods were employed to assess the causal relationship between protein expression and COPD risk: MR Egger, weighted median, inverse variance weighted (IVW), simple mode, and weighted mode, with the IVW method as the focal approach. We also utilized MR-pleiotropy residual sum and outlier (MR-PRESSO) to detect any biased SNPs. Additionally, heterogeneity and pleiotropy tests were performed on the results. A p-value < 0.05 was deemed statistically significant.

## 2.10. Gene Set Variation Analysis (GSVA)

Gene Set Variation Analysis (GSVA) is an unsupervised, non-parametric method that evaluates the enrichment of gene sets based on pathway activity [32]. We utilized the gene set "c2.cp.kegg.symbols" from the Molecular Signatures Database (MSigDB) as a reference for gene pathways. To evaluate the association between ITGB2/HNRNPAB expression and pathway activity, linear regression was performed, and t-values were obtained to indicate the direction and strength of the correlation. The positive t-value indicates a positive correlation between gene expression and pathway activity, while the negative t-value indicates a negative correlation between gene expression and pathway activity. The R package GSVA was employed to score pathway enrichment in COPD and control samples. P-values were calculated to assess statistical significance. Pathways with p < 0.05 were considered significantly correlated. All results were presented with t-value and p-value for complete interpretability.

## 2.11. Preparation of CSE and cell preparation

The CSEr9-050m basic culture medium 1640 was prepared by the principle of negative pressure filtration and placed in a conical flask. Ten Diamond brand cigarettes were placed in a smoke generator. The smoke produced by cigarettes is filtered through the culture medium in a conical flask by vacuum negative pressure filtration. The resulting solution is CSE. The pH value is adjusted to 7.0, and after filtration and sterilization, it is stored at low temperature. This CSE is defined as 100%CSE with a concentration of 1. MLE-12 was purchased from the National Experimental Cell Resource Sharing Platform. The cells were cultured in a 5%CO cell incubator with MLE-12 medium, 10% fetal bovine serum and 1% double antibody. Passage was performed when the cells grew to the logarithmic phase.

## 2.12. Establishment of an animal model of COPD

Mice were randomly divided into a control group and a COPD model group. The COPD model was established in the model group by combining exposure to cigarette smoke with intratracheal instillation of lipopolysaccharide (LPS): daily smoke exposure for 2 hours (30 cigarettes per session), 6 times per week, for 8 consecutive weeks; LPS (50 µg/mouse) was administered via intratracheal instillation on days 1, 14, and 28 of the modeling period. Compared with the control group, the COPD mouse model must meet the following core criteria to be considered successfully established [1–3]: ① Behavioral characteristics: lethargy, dull and yellowish coat; rapid breathing with audible wheezing; significantly reduced activity; decreased appetite; heightened stress response (irritability); ② Body weight changes: Significantly slowed weight gain; body weight at the end of the modeling period was markedly lower than that of the control group; ③ Pathological changes in lung tissue: Demonstrated typical pathological features of emphysema and chronic bronchitis, specifically manifested as alveolar wall rupture, alveolar fusion, and alveolar space enlargement (significant increase in alveolar cross-sectional area); thickened airway walls and narrowed lumens; increased mucus secretion within the airways; ④ Pulmonary function parameters: Measurements using a small-animal spirometer showed that ventilatory function parameters, including forced expiratory volume in 0.1 seconds (FEV0.1), forced expiratory volume in 0.05 seconds (FEV0.05), forced

vital capacity (FVC), and vital capacity (VC), were all significantly reduced. Using impaired lung function and pathological changes associated with emphysema as core indicators can validate the effectiveness of the model.

## 2.13. Lung function test

MeMice were anesthetized via intraperitoneal injection of 1.25% tribromoethanol (Avertin) at a dose of 0.2 mL/10 g body weight. Following the induction of anesthesia, the mice were secured in a supine position on the operating table. The fur on the neck was shaved, and the skin was disinfected with povidone-iodine. A longitudinal incision was made along the midline of the neck, and the subcutaneous tissue and sternocleidomastoid muscle were bluntly dissected. After exposing the trachea, a horizontal incision was made between the tracheal cartilage rings. A tracheal tube was inserted and secured, and the distal end of the tube was connected to a flow-pressure sensor and a small animal ventilator. Using the BUXCO small animal pulmonary function testing system, the following parameters were measured to comprehensively evaluate airway ventilation function: tidal volume (TV), minute ventilation (MV), airway resistance (RL), dynamic lung compliance (Cdyn), forced vital capacity (FVC), forced expiratory volume in 50 ms (FEV50), forced expiratory volume in 100 ms (FEV100), FEV50/FVC, FEV100/FVC, peak expiratory flow (PEF), and maximum mid-expiratory flow (MMEF). The forced expiratory parameters were obtained using the system's built-in negative pressure suction.

## 2.14. HE staining

Sections were dewaxed, rehydrated, and subjected to heat-induced antigen retrieval with EDTA buffer. Following blocking, sections were incubated with primary antibody overnight and fluorophore-conjugated secondary antibody. Nuclei were counterstained with DAPI, autofluorescence was quenched, and sections were mounted with anti-fade medium for fluorescence microscopy.

## 2.15. Real-Time Fluorescent PCR

Total RNA was extracted from a COPD model prepared using TRIzol reagent (EZB-RN4, Suzhou, China) from mouse alveolar epithelial cells (MLE-12). One microgram of total RNA was reverse transcribed into cDNA using a reverse transcription kit (A0010CGQ, Suzhou, China). Mouse β-actin served as the internal control, with primer sequences shown in Table 1. Finally, the SYBR qPCR mixture was used for qRT-PCR analysis on a fluorescence quantitative PCR analyzer (QG-9600, Hangzhou, China). Amplification conditions were as follows: Hot start enzyme activation at 95°C for 5 min; PCR reaction at 95°C for 10 s, 60°C for 30 s, repeated for 40 cycles. Relative mRNA expression levels were calculated using the comparative CT method ($2^{-\Delta\Delta Ct}$).

**Table 1. Primer sequence.**

| Gene name | Primer name | Primer sequence (5'-3') | primer length |
|---|---|---|---|
| Itgb2 | F1 | CAGGAATGCACCAAGTACAAAGT | 23 |
| | R1 | CCTGGTCCAGTGAAGTTCAGC | 21 |
| | F2 | TGCCGCATTCAATGTGACTTT | 21 |
| | R2 | CTTCTTGACGTTGTTGAGGTCAT | 23 |
| Hnrnpab | F1 | TTGACCCTAAAAAGGCTATGGC | 22 |
| | R1 | GAAGCTCAATGGCCTCAATCT | 21 |
| | F2 | TCCCAACACTGGACGATCAAG | 21 |
| | R2 | TAGGGTCAATGACACGACCAT | 21 |
| actin | F1 | GGTCATCACTATTGGCAACGAGC | 23 |
| | R1 | CCAGACAGCACTGTGTTGGCATA | 23 |

## 2.16. Immunohistochemistry

Sections underwent dewaxing, rehydration, and heat-induced antigen retrieval with citrate buffer. After blocking endogenous peroxidase and serum, sections were incubated with primary antibody overnight followed by HRP-conjugated secondary antibody. Staining was visualized with DAB, counterstained with hematoxylin, dehydrated, cleared, and mounted.

## 2.17. Immunofluorescence

Sections were dewaxed, rehydrated, and subjected to heat-induced antigen retrieval with EDTA buffer. Following blocking, sections were incubated with primary antibody overnight and fluorophore-conjugated secondary antibody. Nuclei were counterstained with DAPI, autofluorescence was quenched, and sections were mounted with anti-fade medium for fluorescence microscopy.

## 2.18. Statistical analysis

R software version 4.2.2 was used to conduct all statistical analyses in this study, and a p-value less than 0.05 was considered statistically significant. Data were analyzed using GraphPad Prism 7.0 (GraphPad Software, San Diego, CA, USA). Differences between groups were compared using one-way analysis of variance (ANOVA), assuming data were normally distributed and had equal variances. Statistical significance was set at $P < 0.05$.

## 2.19. Ethics statement

This research was conducted using publically available datasets. No potentially identifiable human images or data is presented in this study.

## 3. Results

### 3.1. Identification of differentially expressed genes (DEGs) for COPD

In this study, DEGs associated with COPD were identified by combining the datasets GSE37768 and GSE38974, and the boxplot and PCA plot after removing batch effects and merging the two datasets were shown in S1 Fig. Using the thresholds of |log FoldChange| ≥ 0.585 and adjusted p-value<0.05, 292 DEGs were identified, comprising 181 upregulated and 111 downregulated genes (Fig 2).

### 3.2. Identification of COPD associated gene modules by Weighted gene co-expression network analysis (WGCNA)

In this study, we employed the WGCNA package (version 1.72–1) in R to construct a co-expression network based on the combined datasets GSE37768 and GSE38974. A soft threshold of 7 was selected to fit a scale-free network with maximum average connectivity, achieving a scale-free R² of 0.95 (Fig 3A). Using the dynamictreecut method, we identified six co-expression modules, each containing over 60 genes (Fig 3B). Among these six significant modules (Fig 3C), the yellow, blue, brown, and gray modules exhibited a positive correlation with COPD, while the green and turquoise modules demonstrated a negative correlation. Notably, only the turquoise module (p = 0.02) met the criterion of p < 0.05, leading us to consider its genes as significantly relevant for further analysis. A significant moderate positive correlation was observed between module membership and gene significance in the turquoise module (r = 0.5, p < 1e-40), confirming its critical role in the potential pathogenesis of COPD (Fig 3D). Ultimately, we identified a total of 753 co-expressed genes.

Subsequently, we intersected the differentially expressed genes from the combined datasets GSE37768 and GSE38974 with the genes from the co-expression modules, resulting in the identification of 112 common genes associated with COPD (S2 Fig).

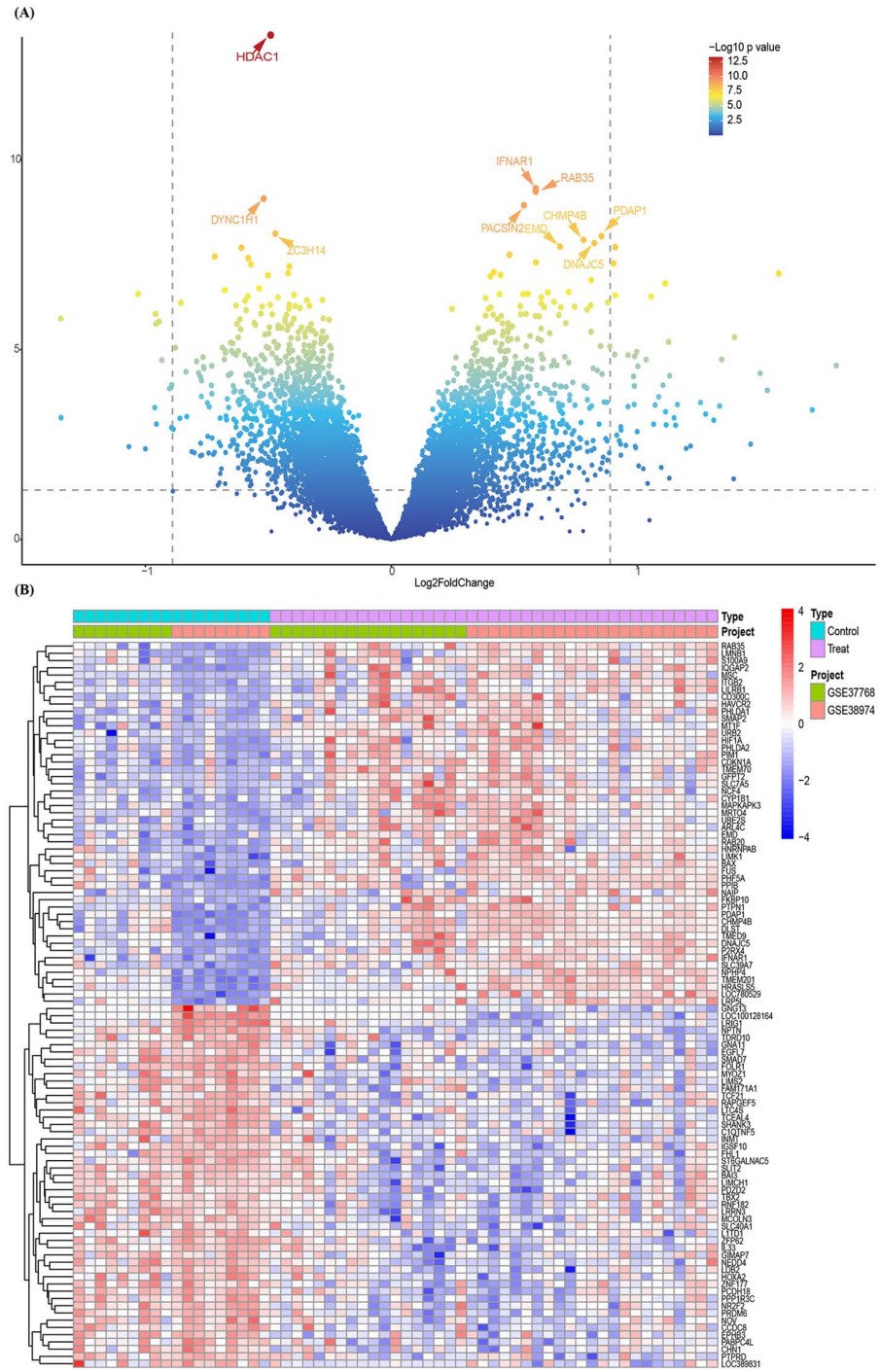

**Fig 2. Differential expression analysis of GSE37768 and GSE38974 combined data setst.** (A) The asymptotic volcano plot of gene expression in the combined dataset of GSE37768 and GSE38974 showed the distribution of all DEGs, with the top ten DEGs specially marked. (B) The heatmap of DEGs in the combined GSE37768 and GSE38974 datasets (n = 292, adjusted p-value < 0.05, |log$_2$FoldChange| ≥ 0.585) illustrated the expression patterns of the top 50 DEGs.

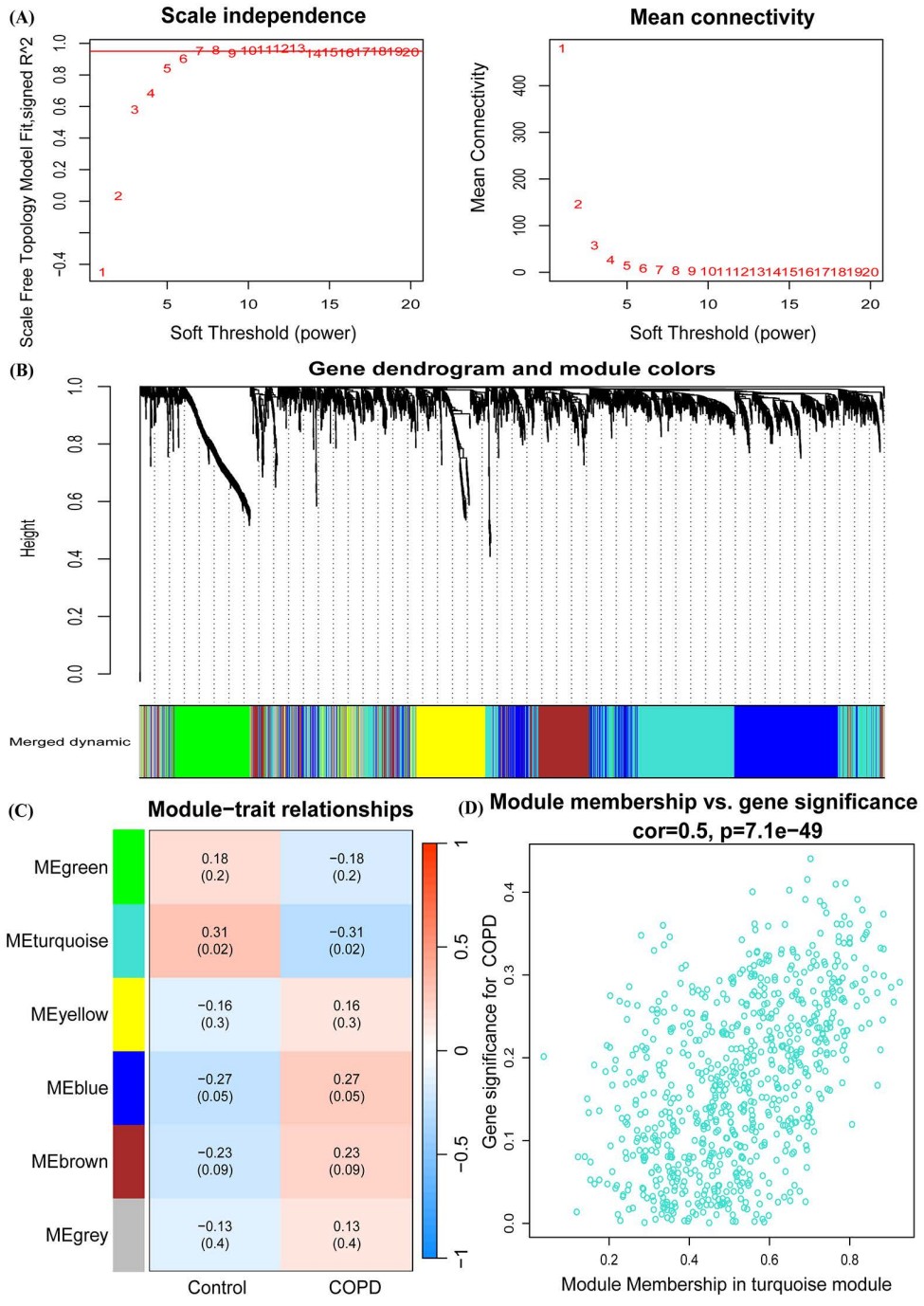

**Fig 3. Identification of significant modules and genes of GSE37768 and GSE38974 combined datasets by WGCNA. (A)** Network topology analysis with different soft thresholds. **(B)** A cluster dendrogram in specific colors reveals six co-expressed gene modules, each with over 60 genes. **(C)** Correlation between disease groupings and gene modules. **(D)** Correlation between module membership and gene significance in the turquoise module.

### 3.3. An integrated framework based on machine learning to predict COPD

To predict COPD using a machine learning-based integrative framework, we employed the combined GSE37768 and GSE38974 datasets as the training cohort, while GSE212331, GSE148004, and GSE1650 served as independent test cohorts for validation. A total of 12 machine learning algorithms were applied, resulting in 113 model combinations (Fig 4A). This approach identified 12 key COPD genes: CD300C, GNG13, HNRNPAB, ITGB2, LDB2, LTC4S, MCOLN3, NAIP, NOV, PDAP1, S100B, and SMAD7 (S1 Table), with corresponding gene expression boxplots shown in Fig 4B. Among these, the combination of glmBoost and plsRglm demonstrated robust diagnostic performance, with an average AUC score of 0.789 and an overall AUC value of 0.989 (Fig 4A, 4C), indicating strong predictive capabilities.

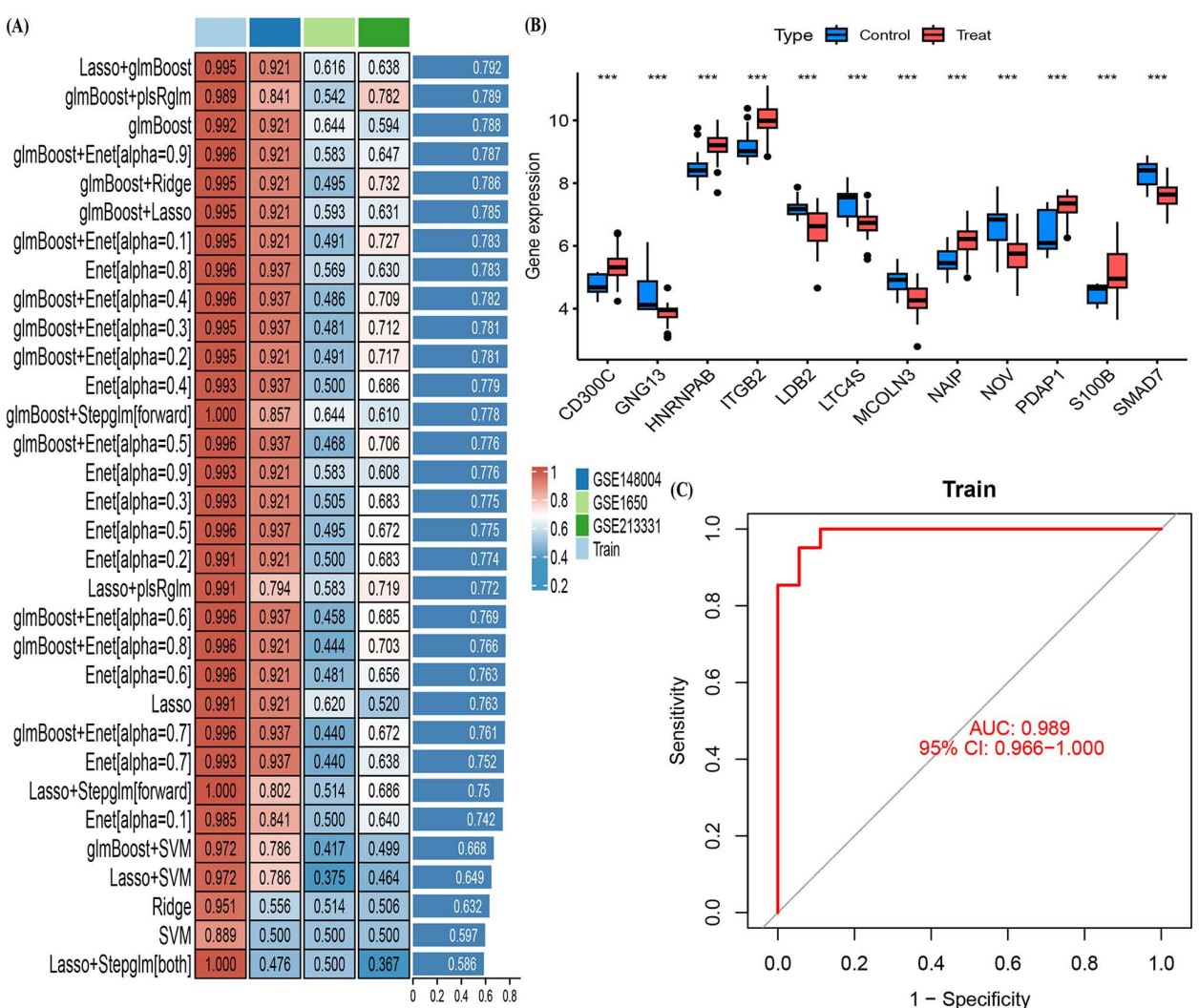

**Fig 4. Performance of Machine Learning Models in Predicting COPD and Expression Patterns of Key Biomarkers. (A)** Comparison of AUC scores for various machine learning model combinations across training and test cohorts (GSE148004, GSE1650, and GSE213331), with the glm-Boost+plsRglm combination achieving the highest AUC of 0.989 (95% CI: 0.966–1.000). **(B)** Expression boxplots of 12 key COPD biomarkers (CD300C, GNG13, HNRNPAB, ITGB2, LDB2, LTC4S, MCOLN3, NAIP, NOV, PDAP1, S100B, SMAD7) between control and COPD groups. These biomarkers were identified as significant in differentiating between the two groups. **(C)** ROC curve showing the diagnostic performance of the best machine learning model (glmBoost+plsRglm) with an AUC of 0.989.

### 3.4. Gene Ontology (GO), KEGG Pathway Analysis

We performed GO and KEGG pathway analyses of COPD-related molecular mechanisms using the "clusterProfiler" package in R. Clustering analysis of the common genes associated with COPD revealed results for three GO categories: biological processes (BP), cellular components (CC), and molecular functions (MF), as well as KEGG pathways. The top 10 terms for each GO category and KEGG pathway are summarized in Table 2. BP terms were primarily enriched in processes such as regulation of long-term synaptic depression, regulation of dendritic cell differentiation, negative regulation of molecular mediator production in immune response, cellular defense response, and tissue migration. In the CC category, actin-based cell projections, filopodia, collagen trimers, cell leading edges, and basement membranes were significantly associated with COPD common genes. MF analysis highlighted amyloid-beta binding, GTPase inhibitor activity, inhibitory MHC class I receptor activity, myosin heavy chain binding, and S100 protein binding as key molecular functions linked to COPD (Fig 5A, 5B).

Notably, the top five pathways identified in the KEGG analysis of this study were the Relaxin signaling pathway, Tryptophan metabolism, Cytoskeletal regulation in muscle cells, AGE-RAGE signaling pathway in diabetic complications, and Glycosphingolipid biosynthesis—ganglio series. These pathway results are illustrated in Fig 5C.

### 3.5. Construction of Transcription factor (TF)-gene regulatory network

Using the JASPAR database's TF binding site profiles, we constructed a TF-gene regulatory network. This network was built based on 12 key COPD diagnostic biomarkers (CD300C, GNG13, HNRNPAB, ITGB2, LDB2, LTC4S, MCOLN3, NAIP, NOV, PDAP1, S100B, and SMAD7), as illustrated in Fig 6. The resulting network comprised 58 nodes and 82 edges, combining 12 seed genes and 46 TFs. Notably, ITGB2 was regulated by nine TFs, while HNRNPAB is controlled by five. Notably, the diamond-shaped transcription factors FOXC1 and NFIC regulated five key genes.

### 3.6. Immune infiltration analysis

We further applied the CIBERSORT algorithm to estimate immune cell infiltration between the COPD and normal groups. The proportion of 22 kinds of immune cells in samples of COPD disease group and control group was shown in Fig 7A. Compared to the control group, the infiltration of Plasma cells, Monocytes, Macrophages M0, Macrophages M1, activated Mast cells, and Neutrophils significantly increased in the COPD group. In contrast, T cells CD8, activated NK cells and Eosinophils showed a significant decrease (Fig 7B).

To further investigate the relationship between gene expression and immune cell infiltration in the COPD microenvironment, we conducted a correlation analysis between multiple genes and immune cell types. The heatmap in Fig 7C illustrates the correlations between various genes, such as CD300C, GNG13, and HNRNPAB, and immune cells including neutrophils, eosinophils, M1 and M2 macrophages, activated NK cells, and memory B cells. Notably, HNRNPAB exhibited a strong negative correlation with plasma cells (correlation coefficient < 0.4) and a positive correlation with naive B cells. Additionally, ITGB2 showed significant negative correlations with T cells CD4 memory resting, NK cells resting, and Mast cells resting, while demonstrating positive correlations with T cells gamma delta and Macrophages M0.

### 3.7. Causal relationship between IGTB2 and HNRNPAB in COPD

We employed Mendelian randomization (MR) to infer causal relationships between 12 COPD biomarkers and COPD risk. Initially, SNPs associated with the proteome of these biomarkers were extracted and aligned with SNPs linked to COPD outcomes. Using the IVW method, we selected results with the highest statistical power ($P < 0.05$) while ensuring consistency in OR directionality. Among these, IGTB2 and HNRNPAB met the criteria, with four SNPs identified for IGTB2 and six for HNRNPAB (Fig 8A). Notably, a genetic predisposition for higher expression of ITGB2 and HNRNPAB significantly increased COPD risk (OR > 1, $P < 0.05$, Fig 8B). A sensitivity analysis was performed to ensure stability of the MR Results, which confirmed no heterogeneity or horizontal pleiotropy in this study.

**Table 2. Top 10 GO category, GO and KEGG pathways and their corresponding P-values and common genes of COPD.**

| ONTOLOGY | ID | Pathway | pvalue | geneID |
|---|---|---|---|---|
| BP | GO:1900452 | regulation of long-term synaptic depression | 4.19E-05 | SHANK3/AGER/LILRB2 |
| BP | GO:2001198 | regulation of dendritic cell differentiation | 4.19E-05 | LILRB1/AGER/LILRB2 |
| BP | GO:0002701 | negative regulation of production of molecular mediator of immune response | 0.000119 | SMAD7/LILRB1/IL33/HMOX1 |
| BP | GO:0006968 | cellular defense response | 0.000177 | CD300C/IL33/RAB23/LILRB2 |
| BP | GO:0090130 | tissue migration | 0.000191 | CYP1B1/CORO1A/SLIT2/FOXF1/HMOX1/ENPP2/CDH13/ACTA2/ACTG2 |
| BP | GO:0072132 | mesenchyme morphogenesis | 0.000221 | FOXF1/ACTA2/ACTG2/TMEM100 |
| BP | GO:0060485 | mesenchyme development | 0.000294 | SMAD7/HNRNPAB/TCF21/FOXF1/ACTA2/ACTG2/TMEM100/EDNRB |
| BP | GO:0072001 | renal system development | 0.000327 | SMAD7/BAX/TCF21/SLIT2/FOXF1/COL4A1/ACTA2/EDNRB |
| BP | GO:0050727 | regulation of inflammatory response | 0.000404 | NAIP/PLA2G2A/IL33/FOXF1/WFDC1/TNFRSF1A/AGER/FFAR2/EDNRB |
| BP | GO:0002704 | negative regulation of leukocyte mediated immunity | 0.000499 | SMAD7/LILRB1/FOXF1/HMOX1 |
| CC | GO:0098858 | actin-based cell projection | 0.000206 | IQGAP2/ARL4C/SLC7A8/AOC3/GPM6A/ACTA2/ACTG2 |
| CC | GO:0030175 | filopodium | 0.000296 | IQGAP2/ARL4C/GPM6A/ACTA2/ACTG2 |
| CC | GO:0005581 | collagen trimer | 0.001203 | C1QTNF5/C1QTNF7/COL4A1/COL4A2 |
| CC | GO:0031252 | cell leading edge | 0.001995 | IQGAP2/LDB2/CORO1A/S100B/SGCE/GABRE/ACTA2/ACTG2 |
| CC | GO:0005604 | basement membrane | 0.002021 | COL4A1/LAD1/COL4A2/ACTA2 |
| CC | GO:0005869 | dynactin complex | 0.003707 | ACTA2/ACTG2 |
| CC | GO:0098644 | complex of collagen trimers | 0.006186 | COL4A1/COL4A2 |
| CC | GO:0046930 | pore complex | 0.008583 | BAX/C8B |
| CC | GO:0062023 | collagen-containing extracellular matrix | 0.008977 | CLEC3B/COL4A1/LAD1/COL4A2/CDH13/ACTA2/OGN |
| CC | GO:0016342 | catenin complex | 0.012067 | CDH3/CDH13 |
| MF | GO:0001540 | amyloid-beta binding | 0.001136 | ITGB2/AGER/LILRB2/BCHE |
| MF | GO:0005095 | GTPase inhibitor activity | 0.00182 | IQGAP2/SLIT2 |
| MF | GO:0032396 | inhibitory MHC class I receptor activity | 0.00182 | LILRB1/LILRB2 |
| MF | GO:0032036 | myosin heavy chain binding | 0.002492 | CORO1A/LIMCH1 |
| MF | GO:0044548 | S100 protein binding | 0.002492 | S100B/AGER |
| MF | GO:0032393 | MHC class I receptor activity | 0.003685 | LILRB1/LILRB2 |
| MF | GO:0008373 | sialyltransferase activity | 0.005611 | ST6GALNAC5/ST6GALNAC3 |
| MF | GO:0042288 | MHC class I protein binding | 0.006151 | LILRB1/LILRB2 |
| MF | GO:0033218 | amide binding | 0.006437 | ITGB2/FKBP10/SLC7A8/AGER/LILRB2/BCHE/EDNRB |
| MF | GO:0017022 | myosin binding | 0.007396 | CORO1A/LIMCH1/MYRIP |
| KEGG | hsa04926 | Relaxin signaling pathway | 0.001443 | GNG13/COL4A1/COL4A2/ACTA2/EDNRB |
| KEGG | hsa00380 | Tryptophan metabolism | 0.002439 | DLST/CYP1B1/INMT |
| KEGG | hsa04820 | Cytoskeleton in muscle cells | 0.003628 | MYOZ1/FHL1/SGCE/COL4A1/COL4A2/ACTG2 |

*(Continued)*

**Table 2.** (Continued)

| ONTOLOGY | ID | Pathway | pvalue | geneID |
|---|---|---|---|---|
| KEGG | hsa04933 | AGE-RAGE signaling pathway in diabetic complications | 0.003983 | BAX/AGER/COL4A1/COL4A2 |
| KEGG | hsa00604 | Glycosphingolipid biosynthesis – ganglio series | 0.004062 | ST6GALNAC5/ST6GALNAC3 |
| KEGG | hsa05146 | Amoebiasis | 0.004273 | ITGB2/COL4A1/COL4A2/C8B |
| KEGG | hsa05417 | Lipid and atherosclerosis | 0.01241 | NCF4/BAX/TNFRSF1A/HSPA2/AGER |
| KEGG | hsa04380 | Osteoclast differentiation | 0.013042 | NCF4/LILRB1/TNFRSF1A/LILRB2 |
| KEGG | hsa04215 | Apoptosis – multiple species | 0.017889 | BAX/TNFRSF1A |
| KEGG | hsa04217 | Necroptosis | 0.019008 | CHMP4B/BAX/IL33/TNFRSF1A |

### 3.8. GSVA pathway analysis of IGTB2 and HNRNPAB

To further explore the potential biological functions of ITGB2, we performed GSVA combined with KEGG pathway enrichment analysis, which quantifies the activity of biological pathways by aggregating the expression levels of genes within each pathway.

Through this integrated analysis, we evaluated the association between ITGB2 gene expression and the activity of various biological pathways. Fig 9A highlighted the significant pathways associated with ITGB2, including immune signaling and metabolic pathways (all p<0.05). Notably, ITGB2 expression was upregulated in the B Cell Receptor Signaling Pathway, Fc epsilon RI Signaling Pathway, and Fc Gamma Receptor Mediated Phagocytosis, indicating its potential role in modulating B cell function and antibody-dependent cellular cytotoxicity. Additionally, ITGB2 expression was associated with significant changes in the activity of multiple metabolic pathways, such as the biosynthesis of unsaturated fatty acids, PPAR signaling, ether lipid metabolism, other glycan degradation, tryptophan metabolism, and sphingolipid metabolism (with positive t-values)(Table 3).

Similarly, for the HNRNPAB gene, we conducted the same GSVA combined with KEGG pathway enrichment analysis to investigate its association with pathway activity. GSVA-based KEGG enrichment analysis revealed distinct differences in pathway activity associated with HNRNPAB expression across immune and metabolic pathways. Fig 9B demonstrated significant immune-related upregulation in pathways such as apoptosis, p53 signaling, Cytosolic DNA Sensing Pathway, and Cytokine-Cytokine Receptor Interaction. Conversely, HNRNPAB was downregulated in pathways like Wnt signaling, ABC transporters, Basal Cell Carcinoma, Melanogenesis, and Hedgehog signaling. In terms of metabolism, HNRNPAB showed upregulation in the sphingolipid metabolism, arginine and proline metabolism, galactose metabolism, and amino sugar and nucleotide sugar metabolism pathways (positive t-values), while displaying a significant downregulation in the propanoate metabolism pathway (all p<0.05) (Table 3).

### 3.9. Cell experiments verified that ITGB2 and HNRNPAB are key genes for COPD

The expression of ITGB2 and HNRNPAB mRNA was detected in mouse lung alveolar epithelial cells (MLE-12) from the COPD model. Compared with the blank group, both ITGB2 and HNRNPAB expression significantly increased in the COPD group (*P*<0.01)(Fig 10A).

Compared with the control group, the IHC results of COPD showed that the average optical density of ITGB2 and the expression of HNRPAB protein were both decreased. At 40x magnification, three random fields of view were selected, and the average optical density values for relevant protein expression were statistically analyzed. Compared with the control group, ITGB2 expression decreased in the COPD group (*P*<0.01), showing a statistically significant difference. The mean optical density value of HNRPA in the COPD group decreased significantly (P<0.01)(Fig 10B). It should be noted that lung tissue sections contain a mixture of cell types, including epithelial cells, immune cells, and stromal cells, whereas MLE-12

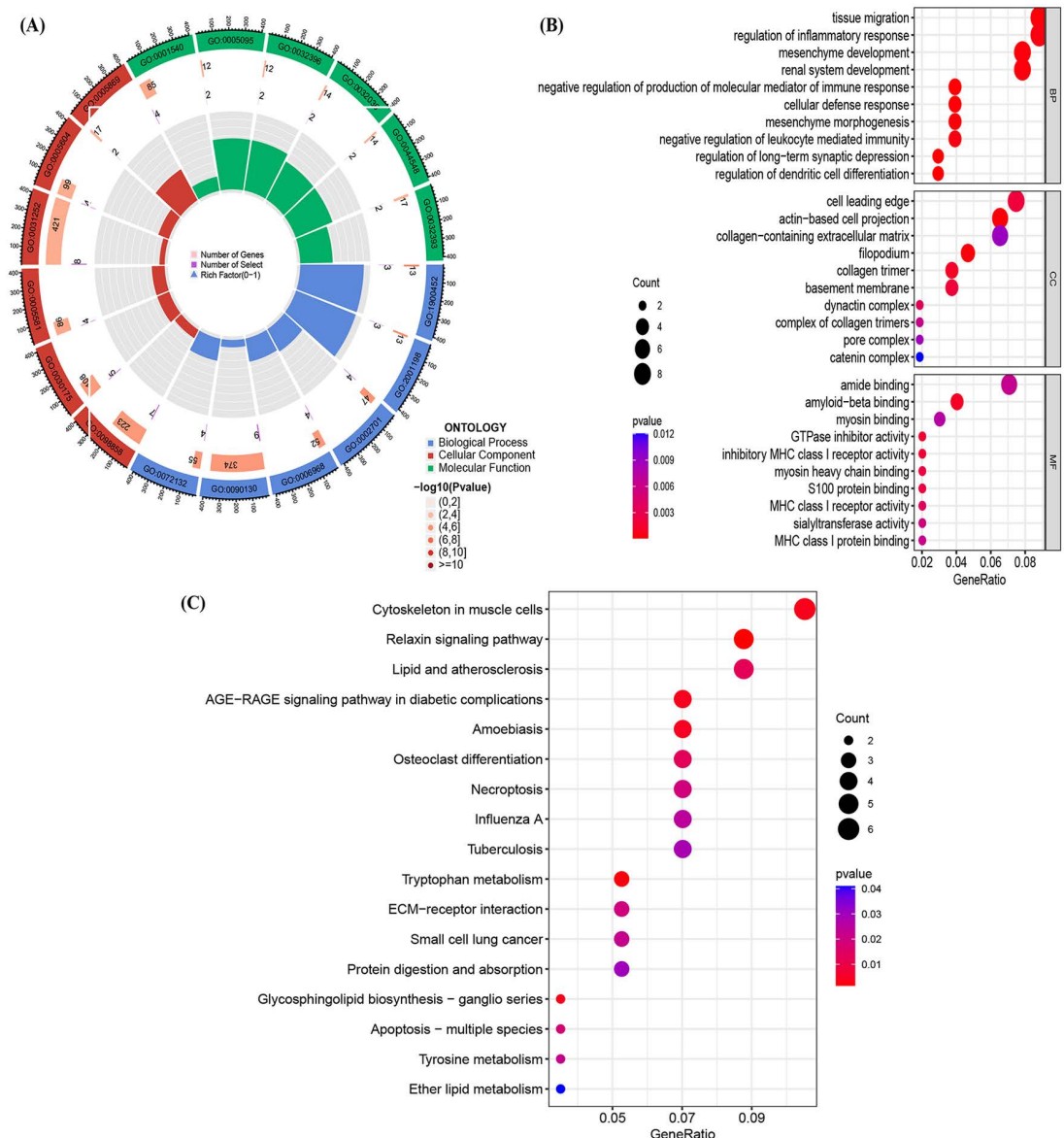

**Fig 5. Functional enrichment analysis of COPD common gene sets. (A)** Ring diagram of GO enrichment analysis. **(B)** Dot plot of GO enrichment analysis. Higher p value indicated a higher number of genes involved in this GO ontology. **(C)** Identification of KEGG enrichment analysis results.

cells represent a pure alveolar epithelial cell line. Therefore, the protein signals in tissue-based analyses reflect the combined contribution of multiple cell populations and may differ from those observed in cultured epithelial cells alone.

Compared with the control group, the mean fluorescence intensity value of ITGB2 protein positive expression decreased in the COPD group ($P < 0.05$). Compared with the control group, the mean fluorescence intensity value of HNRPAB protein positive expression decreased in the COPD group ($P < 0.05$)(Fig 10C). Notably, the mRNA levels of ITGB2 and HNRNPAB were elevated, whereas their protein levels detected by IHC and IF were decreased in the COPD model. This discrepancy may reflect post-transcriptional regulation or differences between the epithelial cell line and mixed lung tissue, and is further addressed in the Discussion.

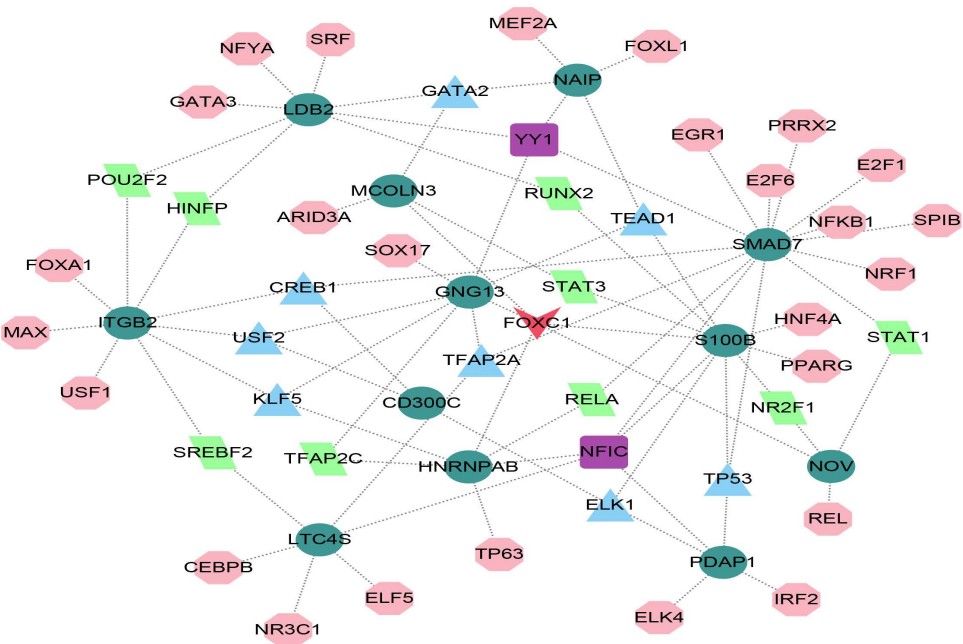

**Fig 6. Transcription factor-gene regulatory network in COPD.** The circular nodes represent key COPD genes, while the adjacent nodes indicated the transcription factors that regulate these genes.

### 3.10. Animal studies have confirmed that ITGB2 and HNRNPAB are key genes in COPD

To confirm successful establishment of the COPD mouse model, we assessed pulmonary function (Table 3) and performed histopathological examination (Fig 11A). Compared with controls, cigarette smoke-exposed mice exhibited typical obstructive ventilatory dysfunction, with significantly increased respiratory frequency (F) and decreased tidal volume (TV, $P<0.001$). Prolonged inspiratory and expiratory times (Te, $P<0.05$) together with reduced peak inspiratory flow (PIF, $P<0.01$) indicated airway obstruction and increased resistance. Minute ventilation (MV) increased compensatorily, but effective alveolar ventilation did not improve. H&E staining (Fig 11B) revealed marked alveolar destruction and enlargement in model mice, confirming emphysematous changes. Collectively, these results demonstrate that chronic cigarette smoke exposure successfully established a COPD mouse model with obstructive ventilatory dysfunction and emphysema. To investigate the expression changes of ITGB2 and HNRPAB in the COPD mouse model, we performed double immunofluorescence staining to detect the protein levels and co-localization of these two molecules in lung tissues from both groups (Fig 11C). In the control group, lung tissues exhibited high expression levels of both ITGB2 and HNRPAB, with a substantial proportion of double-positive cells. In contrast, the COPD group showed significantly reduced expression levels of ITGB2 and HNRPAB, accompanied by a marked decrease in the percentage of double-positive cells. Quantitative analysis (Table 4) further revealed that the mean optical density of ITGB2 decreased from 0.051 in the control group to 0.032 in the model group, with positivity rates declining from 34.00% to 25.02%. Similarly, the mean optical density of HNRPAB dropped from 0.057 to 0.025, with positivity rates decreasing from 61.87% to 34.74%. These results demonstrate that chronic cigarette smoke exposure significantly downregulates both ITGB2 and HNRPAB protein expression and reduces their co-expression in lung tissues of COPD mice, suggesting that these molecules may be involved in the immunoinflammatory regulatory mechanisms associated with COPD pathogenesis.

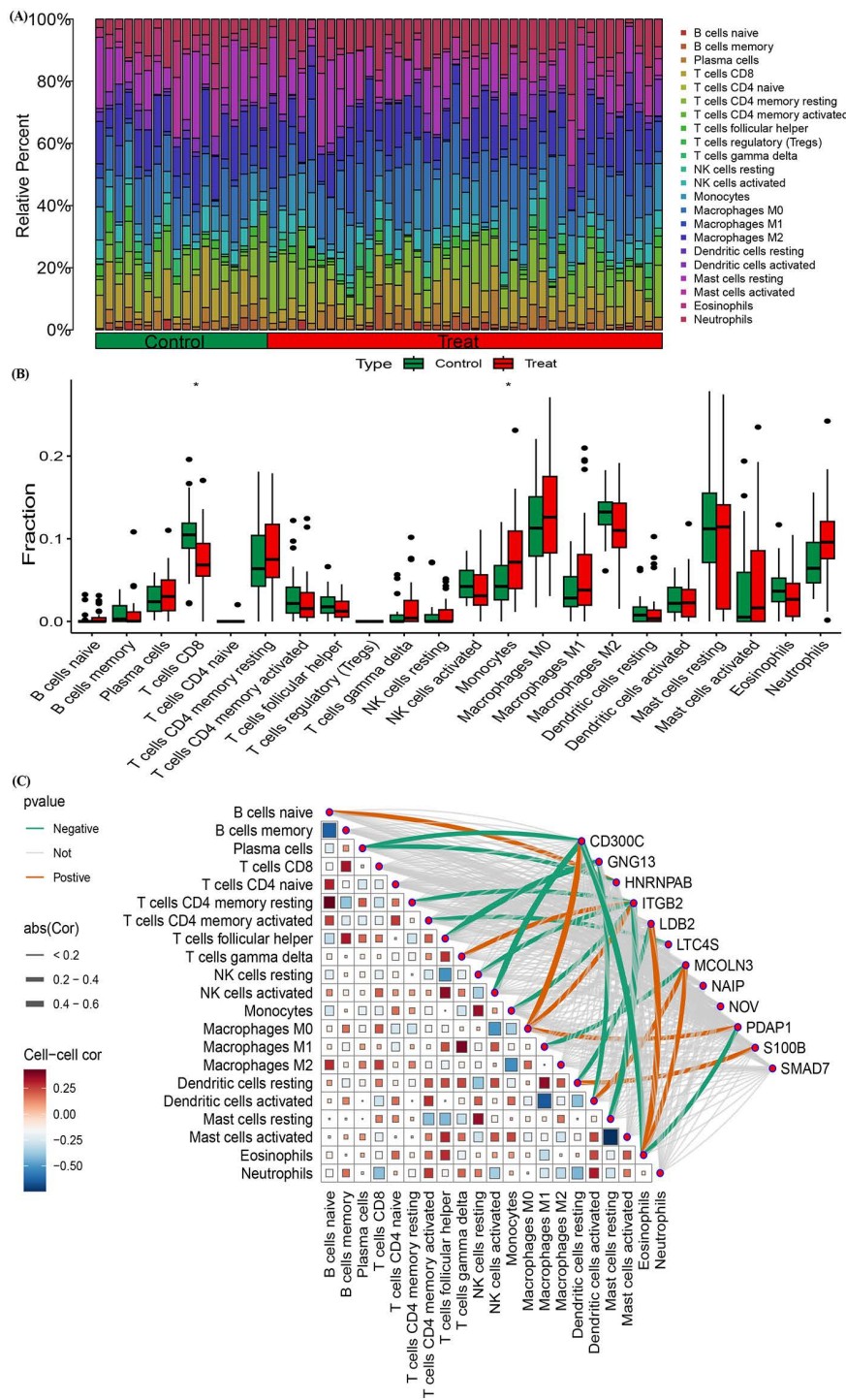

**Fig 7. Immune Cell Infiltration Profiles and Correlation with Key COPD Biomarkers.** (A) Relative proportions of 22 immune cell types in COPD patients and control subjects, as predicted by the CIBERSORT algorithm. (B) Boxplots showing the expression of 12 key COPD biomarkers (CD300C, GNG13, HNRNPAB, ITGB2, LDB2, LTC4S, MCOLN3, NAIP, NOV, PDAP1, S100B, SMAD7) across immune cell types. (C) Heatmap depicting correlations between the key biomarkers and various immune cell types, revealing significant associations between gene expression and immune cell infiltration (correlation coefficients shown).

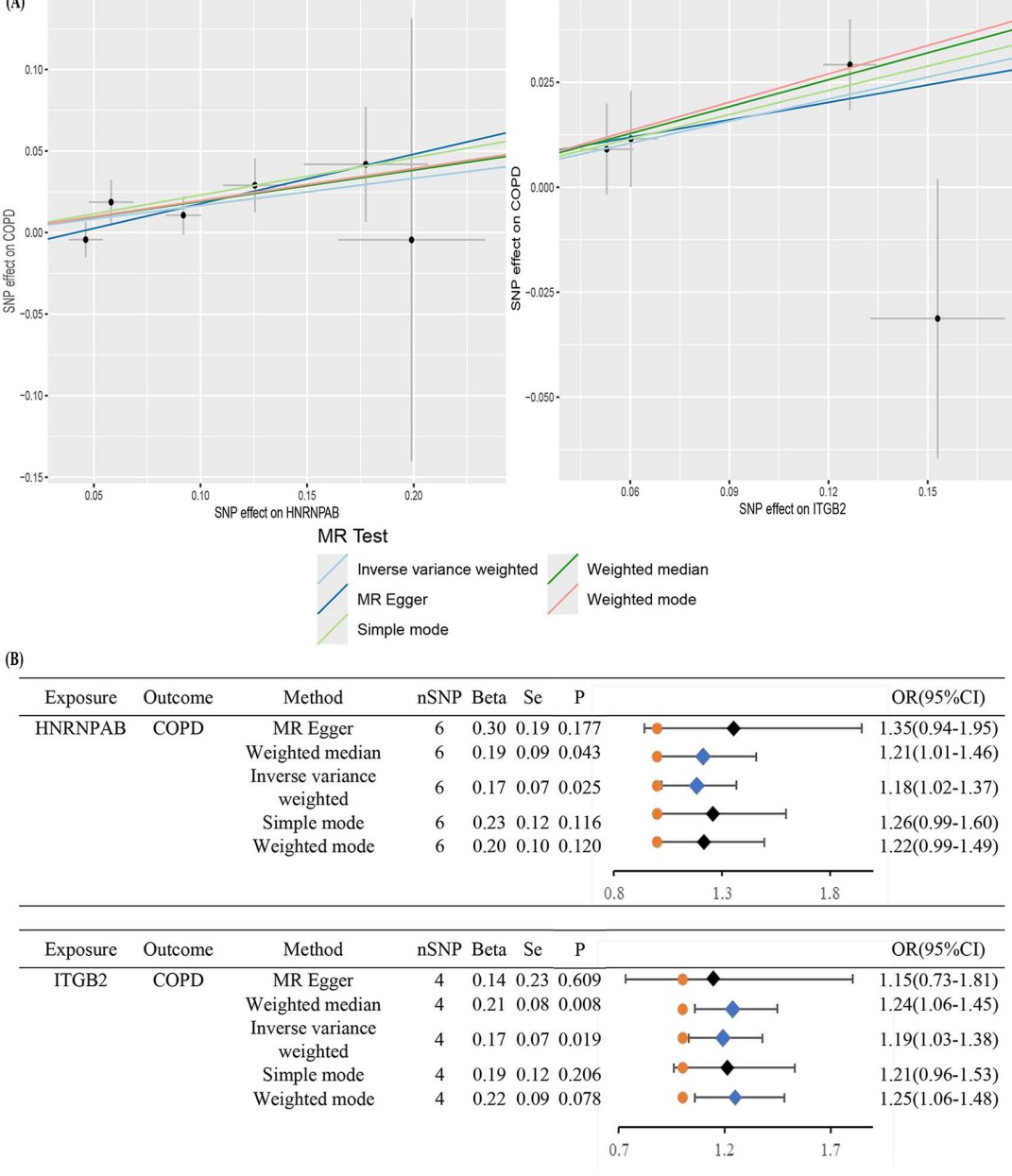

**Fig 8. Mendelian randomization method to identify the causative COPD genes.** (A) Scatter plot of COPD pathogenic genes (HNRNPAB and ITGB2) and COPD. (B) Inverse variance weighting (IVW) was used as the primary method to access the two-way causal relationship between COPD pathogenic genes (HNRNPAB and ITGB2) and COPD.

## 4. Discussion

COPD is a clinically common, gradually worsening respiratory disease, the pathogenesis is chronic inflammation of the respiratory tract and lung parenchyma caused by persistent airflow restriction [1]. The disease is mainly caused by

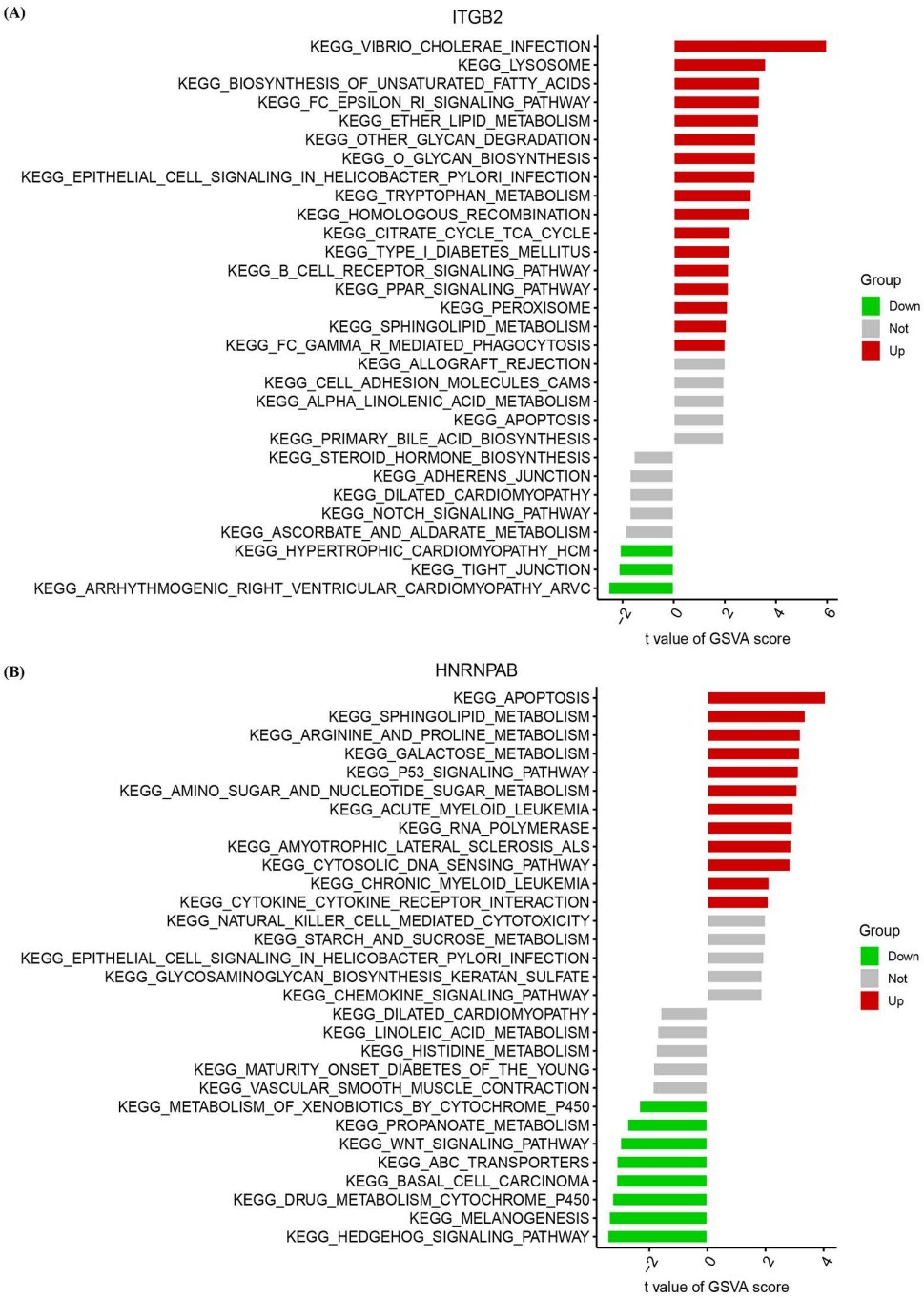

**Fig 9. Gene Set Variation Analysis. (A)** Gene Set Variation Analysis pathway of HNRNPAB. **(B)** Gene Set Variation Analysis pathway of ITGB2.

respiratory conditions such as emphysema and chronic bronchitis, which seriously impair the quality of life of patients and impose a considerable socio-economic burden, and are significantly associated with a large number of morbidity and mortality rates worldwide [33]. The pathogenesis of COPD is multifactorial, involving both environmental factors, notably cigarette smoke exposure [34], and genetic predispositions that contribute to its development and progression [1,35]. Current

**Table 3. The pathway information in the GSVA analysis along with their corresponding t-values and p-values.**

| Gene | Pathway | t | pvalue | Sig |
|------|---------|---|--------|-----|
| ITGB2 | KEGG_ARRHYTHMOGENIC_RIGHT_VENTRICULAR_CARDIOMYOPATHY_ARVC | −2.552753177 | 0.014753 | Down |
| | KEGG_TIGHT_JUNCTION | −2.153202336 | 0.038779 | Down |
| | KEGG_HYPERTROPHIC_CARDIOMYOPATHY_HCM | −2.105256736 | 0.041861 | Down |
| | KEGG_FC_GAMMA_R_MEDIATED_PHAGOCYTOSIS | 2.024502505 | 0.049918 | Up |
| | KEGG_SPHINGOLIPID_METABOLISM | 2.071563212 | 0.045199 | Up |
| | KEGG_PEROXISOME | 2.116637019 | 0.040925 | Up |
| | KEGG_PPAR_SIGNALING_PATHWAY | 2.150218507 | 0.037875 | Up |
| | KEGG_B_CELL_RECEPTOR_SIGNALING_PATHWAY | 2.154755853 | 0.037844 | Up |
| | KEGG_TYPE_I_DIABETES_MELLITUS | 2.195949041 | 0.034111 | Up |
| | KEGG_CITRATE_CYCLE_TCA_CYCLE | 2.216519849 | 0.032848 | Up |
| | KEGG_HOMOLOGOUS_RECOMBINATION | 2.984061327 | 0.005326 | Up |
| | KEGG_TRYPTOPHAN_METABOLISM | 3.045202345 | 0.004183 | Up |
| | KEGG_EPITHELIAL_CELL_SIGNALING_IN_HELICOBACTER_PYLORI_INFECTION | 3.192718283 | 0.002806 | Up |
| | KEGG_O_GLYCAN_BIOSYNTHESIS | 3.209012389 | 0.00268 | Up |
| | KEGG_OTHER_GLYCAN_DEGRADATION | 3.214070784 | 0.002708 | Up |
| | KEGG_ETHER_LIPID_METABOLISM | 3.331326961 | 0.001913 | Up |
| | KEGG_FC_EPSILON_RI_SIGNALING_PATHWAY | 3.367229858 | 0.001773 | Up |
| | KEGG_BIOSYNTHESIS_OF_UNSATURATED_FATTY_ACIDS | 3.374357305 | 0.001738 | Up |
| | KEGG_LYSOSOME | 3.60841657 | 0.000872 | Up |
| | KEGG_VIBRIO_CHOLERAE_INFECTION | 5.99508963 | 5.30E-07 | Up |
| | KEGG_ASCORBATE_AND_ALDARATE_METABOLISM | −1.888858371 | 0.069076 | Not |
| | KEGG_NOTCH_SIGNALING_PATHWAY | −1.722998546 | 0.092813 | Not |
| | KEGG_DILATED_CARDIOMYOPATHY | −1.721914792 | 0.093058 | Not |
| | KEGG_ADHERENS_JUNCTION | −1.719987506 | 0.093395 | Not |
| | KEGG_STEROID_HORMONE_BIOSYNTHESIS | −1.56524262 | 0.12563 | Not |
| | KEGG_PRIMARY_BILE_ACID_BIOSYNTHESIS | 1.955541767 | 0.057881 | Not |
| | KEGG_APOPTOSIS | 1.962979904 | 0.05752 | Not |
| | KEGG_ALPHA_LINOLENIC_ACID_METABOLISM | 1.971901918 | 0.055791 | Not |
| | KEGG_CELL_ADHESION_MOLECULES_CAMS | 1.976809599 | 0.055235 | Not |
| | KEGG_ALLOGRAFT_REJECTION | 2.023394196 | 0.05027 | Not |
| HNRNPAB | KEGG_HEDGEHOG_SIGNALING_PATHWAY | −3.440223211 | 0.001515 | Down |
| | KEGG_MELANOGENESIS | −3.389925313 | 0.001615 | Down |
| | KEGG_DRUG_METABOLISM_CYTOCHROME_P450 | −3.276964928 | 0.00225 | Down |
| | KEGG_BASAL_CELL_CARCINOMA | −3.1391079 | 0.003268 | Down |
| | KEGG_ABC_TRANSPORTERS | −3.126964207 | 0.00336 | Down |
| | KEGG_WNT_SIGNALING_PATHWAY | −3.002001799 | 0.004923 | Down |
| | KEGG_PROPANOATE_METABOLISM | −2.757246818 | 0.009281 | Down |
| | KEGG_METABOLISM_OF_XENOBIOTICS_BY_CYTOCHROME_P450 | −2.349814129 | 0.023959 | Down |
| | KEGG_CYTOKINE_CYTOKINE_RECEPTOR_INTERACTION | 2.102535399 | 0.04243 | Up |
| | KEGG_CHRONIC_MYELOID_LEUKEMIA | 2.133428318 | 0.03955 | Up |
| | KEGG_CYTOSOLIC_DNA_SENSING_PATHWAY | 2.855292296 | 0.006855 | Up |
| | KEGG_AMYOTROPHIC_LATERAL_SCLEROSIS_ALS | 2.885388695 | 0.006491 | Up |
| | KEGG_RNA_POLYMERASE | 2.934693488 | 0.0057 | Up |
| | KEGG_ACUTE_MYELOID_LEUKEMIA | 2.965452994 | 0.005209 | Up |
| | KEGG_AMINO_SUGAR_AND_NUCLEOTIDE_SUGAR_METABOLISM | 3.096517884 | 0.003635 | Up |
| | KEGG_P53_SIGNALING_PATHWAY | 3.141619706 | 0.003391 | Up |

*(Continued)*

**Table 3.** (Continued)

| Gene | Pathway | t | pvalue | Sig |
|---|---|---|---|---|
| | KEGG_GALACTOSE_METABOLISM | 3.186285313 | 0.002855 | Up |
| | KEGG_ARGININE_AND_PROLINE_METABOLISM | 3.212332517 | 0.002707 | Up |
| | KEGG_SPHINGOLIPID_METABOLISM | 3.379059394 | 0.001799 | Up |
| | KEGG_APOPTOSIS | 4.075889903 | 0.000252 | Up |
| | KEGG_VASCULAR_SMOOTH_MUSCLE_CONTRACTION | −1.882521258 | 0.067244 | Not |
| | KEGG_MATURITY_ONSET_DIABETES_OF_THE_YOUNG | −1.867496795 | 0.069755 | Not |
| | KEGG_HISTIDINE_METABOLISM | −1.769770018 | 0.086039 | Not |
| | KEGG_LINOLEIC_ACID_METABOLISM | −1.719942173 | 0.093953 | Not |
| | KEGG_DILATED_CARDIOMYOPATHY | −1.616090374 | 0.114358 | Not |
| | KEGG_CHEMOKINE_SIGNALING_PATHWAY | 1.88992133 | 0.066817 | Not |
| | KEGG_GLYCOSAMINOGLYCAN_BIOSYNTHESIS_KERATAN_SULFATE | 1.890356826 | 0.066209 | Not |
| | KEGG_EPITHELIAL_CELL_SIGNALING_IN_HELICOBACTER_PYLORI_INFECTION | 1.955934786 | 0.058227 | Not |
| | KEGG_STARCH_AND_SUCROSE_METABOLISM | 2.003767934 | 0.052112 | Not |
| | KEGG_NATURAL_KILLER_CELL_MEDIATED_CYTOTOXICITY | 2.006846168 | 0.051735 | Not |

treatment modalities focus on alleviating symptoms and preventing exacerbations [36,37], yet the need for improved early detection and personalized therapeutic strategies remains critical in managing this disease effectively.

The aim of this study was to integrate bioinformatics analytical methods to screen and identify key genetic markers of COPD to address the urgent need for diagnostic methods and effective treatments for COPD. By combining the results of differential gene expression analysis and WGCNA analysis with advanced machine learning ensemble techniques, we have identified a set of 12 diagnostic molecular markers (CD300C, GNG13, HNRNPAB, ITGB2, LDB2, LTC4S, MCOLN3, NAIP, NOV, PDAP1, S100B, and SMAD7) have very high predictive optimality (AUC = 0.989). In a study of blood and sputum transcriptomes, ITGB2 was confirmed to be related to the immune and inflammatory processes of COPD [38]. Liu et al.'s report indicated that PDAP1, through RNA sequencing and in vitro/in vivo experiments, affects the expression of COPD [39], while the other genes have not been emphasized in the existing diagnostic features of COPD. This study constructed a transcription factor gene regulatory network. This network was established based on the prediction of JASPAR gene patterns achieved through NetworkAnalyst 3.0, rather than on co-expression structures or direct experimental evidence of TF binding. FOXC1 and NFIC were predicted to regulate multiple candidate genes including HNRNPAB. Given that NFIC is a known regulator of cell differentiation and apoptosis, and HNRNPAB is an RNA-binding protein controlling mRNA stability and splicing, their predicted connection suggests a novel two-layer regulatory mechanism in COPD [40]. Thus, the network should be interpreted as a computational framework for identifying putative upstream transcriptional regulators of the 12 COPD-related biomarkers.

The functional enrichment analyses, particularly via GO and KEGG, illuminate critical biological processes and pathways implicated in COPD. The results of our study highlight the significant involvement of the Relaxin signaling pathway in the pathophysiology of COPD, particularly in muscle cells. Relaxin is a peptide hormone that plays a crucial role in regulating extracellular matrix remodeling and muscle cell function [41], its signaling cascade activates various intracellular pathways [42], promotes muscle relaxation and inhibits fibrosis, and is closely related to muscle atrophy and dysfunction caused by COPD [43]. Furthermore, our findings suggest that the cytoskeletal regulatory pathway is involved in the disease process of COPD. The cytoskeleton is essential for maintaining cellular structure and facilitating intracellular transport [44]. In COPD, remodeling of cytoskeletal components such as actin and tubulin may lead to observed muscle atrophy and reduced contractility [45]. Collectively, these pathways present promising avenues for further research and potential therapeutic strategies in the management of COPD-related muscle dysfunction.

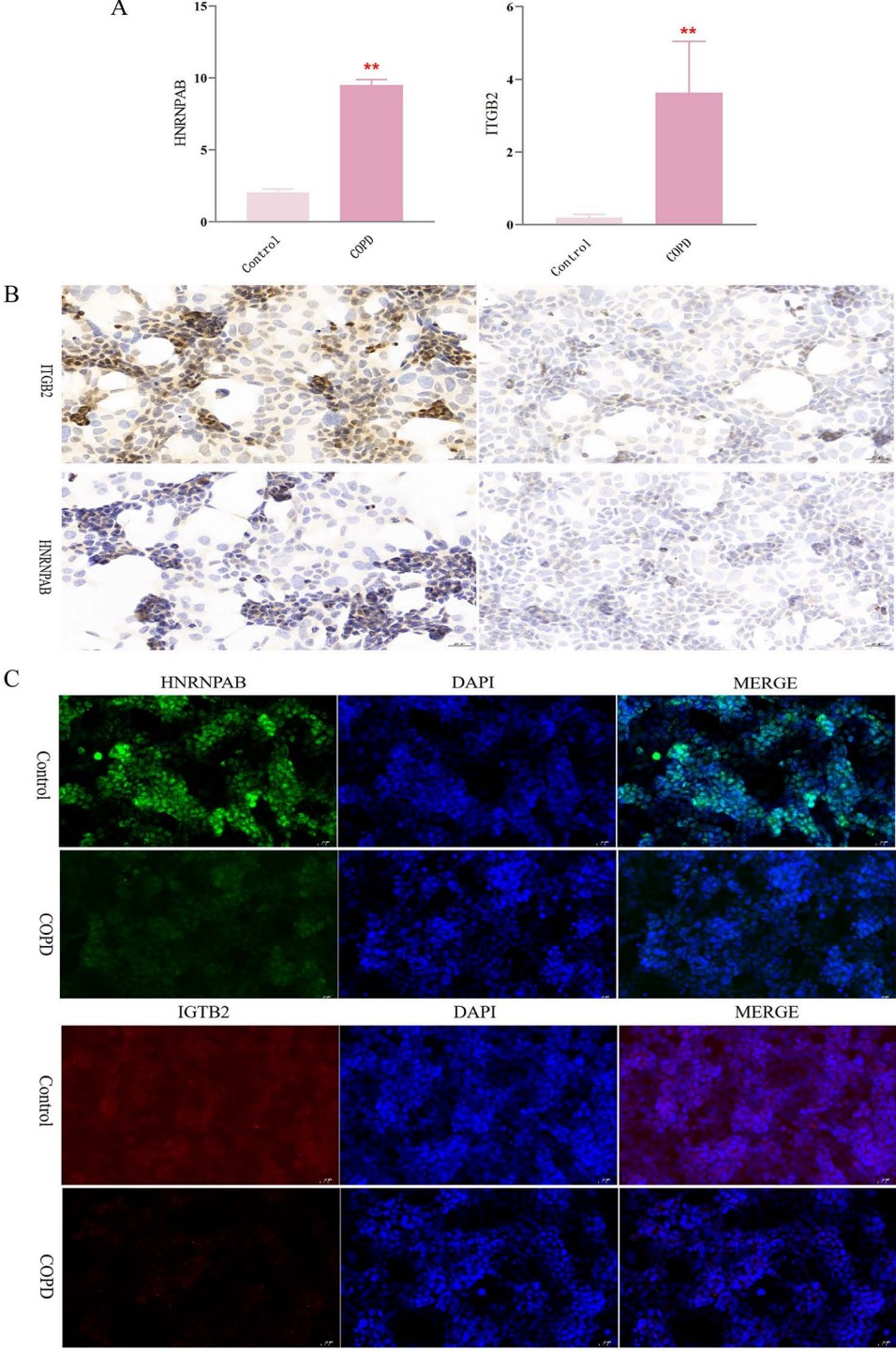

**Fig 10. The expression of HNRNPAB and ITGB2 in the cell model in the Control group and the COPD group. (A)** Quantitative real-time PCR analysis of HNRNPAB and ITGB2 mRNA expression levels in control and compound-treated groups. Data are presented as mean±SEM; *p<0.05, **p<0.01 vs. Control. **(B)** Immunohistochemical (IHC) staining showed ITGB2 and HNRNPAB in tissue sections from control and COPD groups. **(C)** Immunofluorescence (IF) staining showed subcellular localization of HNRNPAB and ITGB2 in control and compound-treated cells. Nuclei were counterstained with DAPI. Merged images illustrate colocalization.

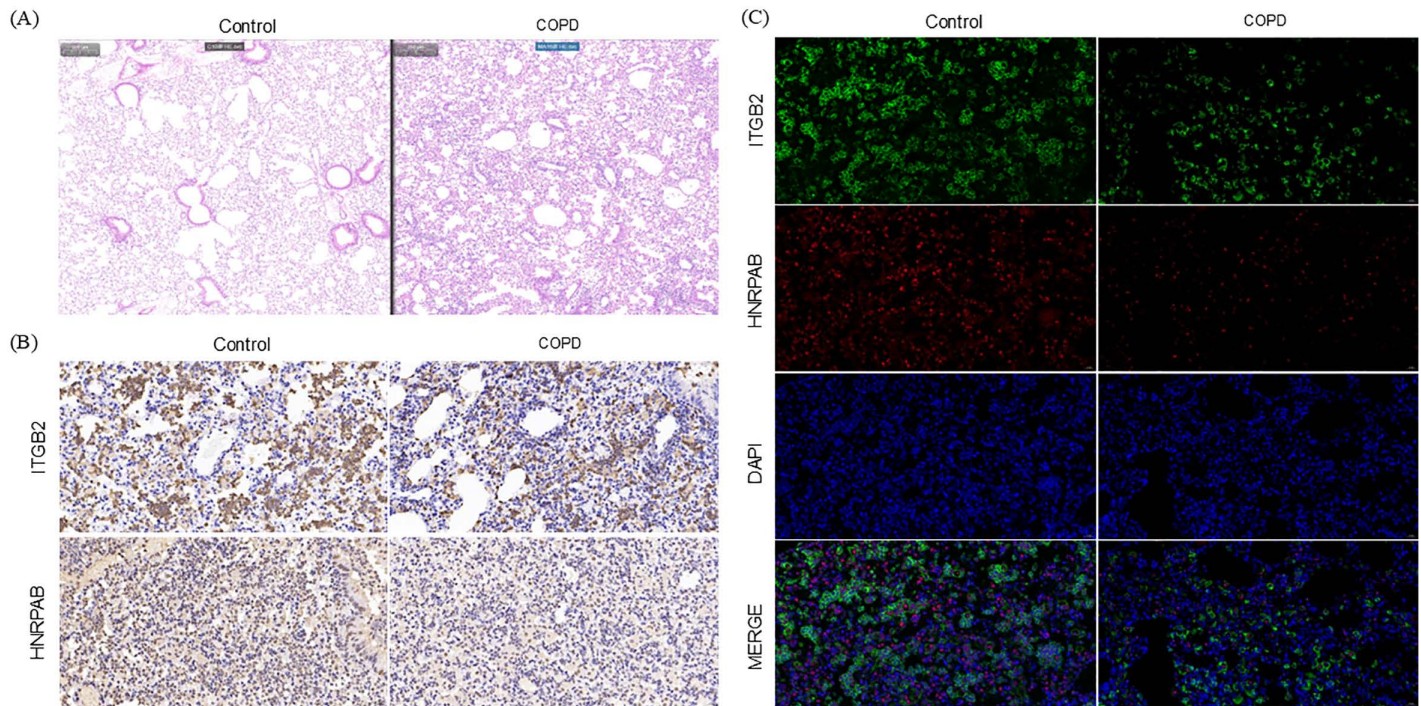

**Fig 11. The expression levels of HNRNPAB and ITGB2 in the animal model in the control group and the chronic obstructive pulmonary disease group. (A)**The result of H&E. **(B)** IHC staining showed ITGB2 and HNRNPAB in tissue sections from control and COPD groups. **(C)** IF staining revealed the subcellular localization of HNRNPAB and ITGB2 in the animal samples of the control group and the COPD group.

**Table 4. Pulmonary function data.**

| Group | Ti (s) | Te (s) | F(breaths/min) | TV (mL) | MV (mL/min) | EF50 (mL/s) | PIF (mL/s) |
|---|---|---|---|---|---|---|---|
| Control | 0.26303±0.00544 | 0.10693±0.00615 | 140.472±0.348 | 0.21433±0.00860 | 23.203±0.0545 | 0.68772±0.0107 | 1.4810±0.1003 |
| COPD | 0.31391±0.01260 | 0.13335±0.00720 | 199.61±9.25 | 0.13383±0.00286 | 31.628±0.854 | 0.91252±0.00551 | 0.8585±0.0731 |

In addition, the immune infiltration analysis in COPD samples revealed significant alterations in the composition of immune cell populations. Among the various immune cells, the increased presence of plasma cells warrants particular attention. Plasma cells, as terminally differentiated B cells, are crucial for antibody production [46]. Their elevated infiltration in COPD suggests a heightened humoral immune response, potentially contributing to the chronic inflammation characteristic of this disease. Monocytes and macrophages also exhibited significant changes in their infiltration patterns. The increased presence of M0 and M1 macrophages indicates a shift towards a pro-inflammatory phenotype, which is consistent with the chronic inflammatory state observed in COPD [47,48]. Furthermore, the observed reduction in CD8＋T cells and activated NK cells raises important questions regarding the adaptive immune response in COPD. CD8＋T cells are deeply involved in clearing viral infections and modulating immune responses [49]. Their diminished infiltration may suggest an impaired ability to control inflammation and infection, potentially leading to exacerbations in COPD patients [50]. Overall, these findings underscore the complex interplay of immune cells in COPD.

The findings from Mendelian randomization analysis provide compelling evidence for the causal relationship between specific genes (ITGB2 and HNRNPAB) and COPD risk. The observed association between elevated expression levels of

these genes and increased susceptibility to COPD suggesting their preliminary potential as exploratory diagnostic markers and research-oriented therapeutic targets.

Integrin beta-2 (ITGB2) is a crucial component of the integrin family, which plays a significant role in cell adhesion, migration, and signaling [51–53]. It is primarily expressed on leukocytes and is essential for their interaction with the endothelium during immune responses. Recent studies have highlighted the role of ITGB2 in COPD, which is associated with the pathogenesis of airway inflammation and cellular immunity [54]. ITGB2's role in modulating immune responses positions it as a target for therapeutic interventions aimed at mitigating inflammation in respiratory diseases [54].

Heterogeneous nuclear ribonucleoprotein A/B (HNRNPAB) is a multifunctional RNA binding protein involved in RNA metabolism such as splicing, transport and stability [55,56]. Scientific studies have shown that HNRNPAB is involved in the regulation of gene expression in the context of COPD. Specifically, it has been shown to have RNA activity in the COPD disease state [57], with which it induces epithelial-mesenchymal transition (EMT), as a potential factor in airway disease. The dysregulation of HNRNPAB in COPD patients suggests that it may contribute to the altered gene expression profiles observed in this disease, thereby providing insights into the molecular mechanisms underlying COPD pathogenesis.

By combining bioinformatics screening, machine learning, and Mendelian randomization analysis, ITGB2 and HNRNPAB were identified as genes highly associated with the pathological risk of chronic obstructive pulmonary disease (COPD). In subsequent experimental validations, we observed that the mRNA levels of these two genes were significantly upregulated in the alveolar epithelial cell line (MLE-12) of COPD model mice. However, their protein expressions in the cell lines and lung tissue sections showed a downward trend as assessed by immunohistochemistry and immunofluorescence. This significant inconsistency between the transcriptomic and proteomic data does not simply negate the results of the computational screening, but may reveal a complex and intricate multi-level gene expression regulatory mechanism during the COPD process.

This inconsistency between mRNA and protein levels suggests the presence of active post-transcriptional regulation in COPD. Inflammatory and oxidative stress may increase ITGB2 and HNRNPAB transcription, while their translation or protein stability is inhibited, possibly via miRNA-mediated suppression, ubiquitin-proteasome degradation, or autophagy-lysosomal pathways. Such a pattern supports a "high transcription–high degradation" equilibrium in the COPD microenvironment.

Disease stage and cell-type specificity may also contribute to these differences. mRNA upregulation could represent an acute stress response, whereas reduced protein levels may reflect late-stage decompensation. Immunostaining signals also reflect mixed cell populations, including epithelial cells, immune cells, and stromal cells, which may differ greatly from purified cultured cells. Methodological differences may also contribute: qPCR sensitively quantifies mRNA, while immunohistochemistry and immunofluorescence are semi-quantitative and influenced by antibody efficiency, antigen retrieval, and subcellular protein redistribution. Despite these discrepancies, our findings support ITGB2 and HNRNPAB as candidate molecules in COPD pathogenesis. The uncoupling of transcription and translation provides a novel perspective regarding epithelial dysfunction, cell adhesion, and RNA regulatory network disorders in COPD. Future studies using Western blotting, protein degradation assays, RNA immunoprecipitation, and gene manipulation will help clarify the precise mechanisms of these molecules.

While our study establishes a theoretical foundation and research paradigm, several limitations are inherent in our model. The data utilized in this analysis were sourced from the GEO repository, which introduces variability in the quality and reliability of the statistical metrics. To mitigate this, we strategically selected GSE37768 and GSE38974 as our primary datasets, and validated our model using GSE212331, GSE148004, and GSE1650, given their well-defined cohort stratification. It is worth noting that the predictive model obtained by integrating machine learning in this study has an AUC of 0.989, which may involve overfitting or poor specificity of the GEO sample dataset. Furthermore, in the evolving landscape of multi-omics research, it is imperative that our findings be integrated with single-cell sequencing data to enhance

the comprehensiveness of the analysis. Ultimately, the mechanistic roles and interplay between the two identified COPD susceptibility genes warrant further investigation to elucidate their pathogenic significance in COPD.

## 5. Conclusion

In conclusion, this study utilized bioinformatics and Mendelian randomization methods to identify HNRPAB and ITGB2 as promising candidate molecular targets for COPD. Preliminary verification in mouse cell and lung tissue models supports their potential as exploratory diagnostic biomarkers, pending further prospective human cohort validation and comparison with standard clinical indices.

## Supporting information

**S1 File. Supplementary Explanation of Machine Methods.**
(DOCX)

**S1 Fig. Boxplots and PCA plots of the removal batch effect for the combined gene set of GSE37768 and GSE38974.**
(TIF)

**S2 Fig. Venn plots of differentially expressed genes of GSE37768 and GSE38974 intersected with genes from co-expression modules.**
(TIF)

**S1 Table. Relevant functions of 12 COPD diagnostic molecular markers screened by machine learning integrated framework.**
(DOCX)

## Acknowledgments

We thank Dr. Guangli Sun, Chief Physician for her critical comments on the manuscript.

## Author contributions

**Conceptualization:** Fengjun Zhang, Hui Li.

**Data curation:** Fan Wu, Dexian Xian.

**Formal analysis:** Fengjun Zhang.

**Funding acquisition:** Wei Zhang.

**Investigation:** Wenchang Xu.

**Methodology:** Fengjun Zhang, Hui Li.

**Project administration:** Xiaodan Liu, Wei Zhang.

**Resources:** Yuchen He.

**Software:** Feng Chen.

**Supervision:** Wei Zhang.

**Validation:** Xiaodan Liu.

**Visualization:** Hui Li.

**Writing – original draft:** Fengjun Zhang, Hui Li.

**Writing – review & editing:** Xiaodan Liu, Wei Zhang.

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
