## [Decision Letter · Decision Letter 0]

25 Nov 2025

PONE-D-25-52871Identifying and Validating ITGB2 and HNRNPAB as diagnostic biomarkers in chronic obstructive pulmonary disease Using Bioinformatics and Integrated Machine Learning MethodsPLOS ONE

Dear Dr. Zhang,

Thank you for submitting your manuscript to PLOS One. Firstly, we would like to apologize for the delay in processing your manuscript. It has been exceptionally difficult to secure reviewers to evaluate your study. We have now received one completed review, which is available below. The reviewer has raised scientific concerns about the study that need to be addressed in a revision.

Please note that we have only been able to secure a single reviewer to assess your manuscript. We are issuing a decision on your manuscript at this point to prevent further delays in the evaluation of your manuscript. Please be aware that the editor who handles your revised manuscript might find it necessary to invite additional reviewers to assess this work once the revised manuscript is submitted. However, we will aim to proceed on the basis of this single review if possible.

A rebuttal letter that responds to each point raised by the academic editor and reviewer(s). You should upload this letter as a separate file labeled ‘Response to Reviewers’.A marked-up copy of your manuscript that highlights changes made to the original version. You should upload this as a separate file labeled ‘Revised Manuscript with Track Changes’.An unmarked version of your revised paper without tracked changes. You should upload this as a separate file labeled ‘Manuscript’.

We look forward to receiving your revised manuscript.

Kind regards,

Miquel Vall-llosera Camps

Senior Staff Editor

PLOS One

3. Please include a caption for figures 2-10.

Reviewers' comments:

Reviewer's Responses to Questions

**Comments to the Author**

1. Is the manuscript technically sound, and do the data support the conclusions?

Reviewer #1: Yes

2. Has the statistical analysis been performed appropriately and rigorously? 

Reviewer #1: Yes

3. Have the authors made all data underlying the findings in their manuscript fully available?

Reviewer #1: Yes

4. Is the manuscript presented in an intelligible fashion and written in standard English?

Reviewer #1: Yes

5. Review Comments to the Author

Reviewer #1: The manuscript by Zhang et al.” Identifying and Validating ITGB2 and HNRNPAB as diagnostic biomarkers in chronic obstructive pulmonary disease Using Bioinformatics and Integrated Machine Learning Methods" identifies COPD biomarkers from multiple gene expression datasets deposited in the GEO database, by combining several bioinformatics approaches including WGCNA, immune infiltration profiling, transcription factor and regulatory network analysis, enrichment analyses, MR, GSVA, PCR validation, and integrated machine-learning methods. Two of the identified biomarkers, ITGB2 and HNRNPAB, were further validated through PCR, IHC and IF experiments in a mouse model.

The experiments are well designed, appropriately analyzed, and the methodology is clearly described throughout the manuscript. The results are of interest to the field and, therefore, I recommend publication after minor revisions aimed at further improving the clarity and completeness of the work.

Minor comments:

1. Methods and Materials (line 100):

The sentence “…and GSE1650 consists of 18 COPD patient samples and 12 healthy control samples [17]…” is incomplete. Please clarify for what purpose this dataset was used in the study.

2. Identification of DEGs in COPD (lines 109–110):

The authors state that genes were defined as DEGs if they met the criteria of |log Fold Change| ≥ 0.585 and adjusted p-value < 0.05.

Why was the threshold set at 0.585 specifically, rather than 0.586, 0.584, or a more conventional 0.5? Please provide a justification for this choice.

3. Discussion section:

The results obtained from the validation of ITGB2 and HNRNPAB in the mouse model should be revisited and discussed in this section, clearly justifying their purpose and contribution to the interpretation and validation of the findings.

In addition, the sentence “The transcriptional factors FOXC1 and NFIC play an important role in the regulation of COPD” (lines 446–447) should be better integrated with the other proteins/pathways identified and discussed in the manuscript.

6. PLOS authors have the option to publish the peer review history of their article (what does this mean?). If published, this will include your full peer review and any attached files.

Reviewer #1: No

---

## [Author Response · Author response to Decision Letter 1]

12 Dec 2025

Dear Miquel Vall-llosera Camps and Reviewer:

Thank you for your letter and for the reviewers’ comments concerning our manuscript entitled “Identifying and Validating ITGB2 and HNRNPAB as diagnostic biomarkers in chronic obstructive pulmonary disease Using Bioinformatics and Integrated Machine Learning Methods”. (Manuscript Number: PONE-D-25-52871). Those comments are all valuable and very helpful for revising and improving our paper, as well as the important guiding significance to our research. We have studied comments carefully and have made corrections which we hope meet with approval. Revised portions are marked with a yellow background in the revised version. The main corrections in the paper and the responds to the reviewer’s comments are as flowing:

Responses to reviewer’s comments: (original comments by reviewers are in blue color). Furthermore, we have added the legend explanations as per the requirements, which are located at the bottom of the manuscript (lines 817 to 866).

Reviewer #1:

1. Response to comment: The manuscript by Zhang et al.” Identifying and Validating ITGB2 and HNRNPAB as diagnostic biomarkers in chronic obstructive pulmonary disease Using Bioinformatics and Integrated Machine Learning Methods" identifies COPD biomarkers from multiple gene expression datasets deposited in the GEO database, by combining several bioinformatics approaches including WGCNA, immune infiltration profiling, transcription factor and regulatory network analysis, enrichment analyses, MR, GSVA, PCR validation, and integrated machine-learning methods. Two of the identified biomarkers, ITGB2 and HNRNPAB, were further validated through PCR, IHC and IF experiments in a mouse model.

The experiments are well designed, appropriately analyzed, and the methodology is clearly described throughout the manuscript. The results are of interest to the field and, therefore, I recommend publication after minor revisions aimed at further improving the clarity and completeness of the work.

Minor comments:

1. Methods and Materials (line 100):

The sentence “…and GSE1650 consists of 18 COPD patient samples and 12 healthy control samples [17]…” is incomplete. Please clarify for what purpose this dataset was used in the study.

Response: Thank you very much for the Reviewer’s comments. We have already provided supplementary explanations for the original text. In this study, we first conducted difference analysis and WGCNA analysis on the GSE37768 and GSE38974 datasets merged by the sva algorithm. The intersected differential gene inputs were further analyzed by 113 machine learning models. The GSE212331, GSE148004 and GSE1650 datasets were used as independent queues to verify the stability of the above-mentioned machine learning models.

2. Response to comment: Identification of DEGs in COPD (lines 109–110):

The authors state that genes were defined as DEGs if they met the criteria of |log Fold Change| ≥ 0.585 and adjusted p-value < 0.05. Why was the threshold set at 0.585 specifically, rather than 0.586, 0.584, or a more conventional 0.5? Please provide a justification for this choice.

Response: Special thanks to you for your exact and professional comments. We set the log2 FoldChange to 0.585 based on the principle of logarithmic transformation. The location of the difference multiple is "gene expression level of the treatment group/gene expression level of the control group" (up-regulated genes) or "gene expression level of the control group/gene expression level of the treatment group" (down-regulated genes). Since the expression levels of RNA-seq data (such as FPKM and TPM) usually follow a lognormal distribution, logarithmic transformation to the base 2 (log2 (FC)) is commonly used in bioinformatics analysis to make the data meet the normality hypothesis and facilitate subsequent statistical tests. When FC=1.5, the corresponding log2 (FC)=log2 (1.5)≈0.58496. To simplify the calculation and ensure the uniformity of the threshold, this study rounded it to 0.585. In previous studies on respiratory diseases, the setting of this threshold was also used by other researchers [1]. This conversion logic is a standard operation for screening differentially expressed genes in bioinformatics analysis, ensuring the mathematical rigor of the threshold setting. Thank you for your review. We have revised the original text, which ensures the accuracy of the conclusion.

Reference:

1. Wu Z, Chen H, Ke S, Mo L, Qiu M, Zhu G, et al. Identifying potential biomarkers of idiopathic pulmonary fibrosis through machine learning analysis. Sci Rep. 2023;13(1):16559. doi: 10.1038/s41598-023-43834-z. PubMed PMID: 37783761; PubMed Central PMCID: PMCQ2.

3. Response to comment: Discussion section:

The results obtained from the validation of ITGB2 and HNRNPAB in the mouse model should be revisited and discussed in this section, clearly justifying their purpose and contribution to the interpretation and validation of the findings.

Response: Thank you sincerely for your meticulous review and highly constructive suggestions. Based on your suggestions, we have made extensive revisions to the discussion section of the paper. We have systematically analyzed a variety of biological and technical reasons that may lead to inconsistent expression trends at mRNA and protein levels. We emphasize that this discovery has not weakened the value of ITGB2 and HNRNPAB as important biomarkers for COPD. On the contrary, it suggests that in the complex pathological environment of COPD, gene expression regulation has multi-level and dynamic characteristics. Computational biology screening has helped us identify these two genes strongly associated with diseases, and the "contradictions" revealed by wet experimental verification precisely point to the possible existence of novel biological mechanisms (such as active protein turnover or translation inhibition), which in itself is a valuable discovery. During the revision, we also candidly admitted the limitations of the current research, that is, the initial verification failed to fully unify the expression level. We have taken this as the starting point for future research. At the end of the discussion, we proposed several hypotheses and experimental directions that can be further verified (such as detecting protein degradation pathways and conducting functional recovery experiments, etc.) to promote subsequent exploration in this field.

We believe that the revised discussion section not only directly addresses your concerns but also enhances the scientific depth of the paper, transforming seemingly contradictory data into an opportunity to tell a richer biological story.

Thank you again for your valuable suggestions. We hope our revisions can earn your approval of the paper.

4. Response to comment:

In addition, the sentence “The transcriptional factors FOXC1 and NFIC play an important role in the regulation of COPD” (lines 446–447) should be better integrated with the other proteins/pathways identified and discussed in the manuscript.

Response: We thank the reviewer for the critical suggestion. In response, we have thoroughly revised the discussion to integrate FOXC1 and NFIC into our molecular narrative. We moved beyond stating their“important role”to highlighting a concrete regulatory relationship predicted by our network analysis: NFIC directly targets HNRNPAB. This axis is presented as a central piece connecting upstream regulation (TF network), core biomarkers (HNRNPAB’s prioritization by MR), and downstream pathology. This creates a cohesive story from discovery to mechanistic insight.

We believe this revision directly addresses your comment by deeply embedding the transcription factors within a specific and plausible biological framework derived from our own data.

---

## [Decision Letter · Decision Letter 1]

3 Feb 2026

PONE-D-25-52871R1Identifying and Validating ITGB2 and HNRNPAB as diagnostic biomarkers in chronic obstructive pulmonary disease Using Bioinformatics and Integrated Machine Learning MethodsPLOS One

Dear Dr. Zhang,

Thank you for submitting your manuscript to PLOS ONE. After careful consideration, we feel that it has merit but does not fully meet PLOS ONE’s publication criteria as it currently stands. Therefore, we invite you to submit a revised version of the manuscript that addresses the points raised during the review process.

We look forward to receiving your revised manuscript.

Kind regards,

Tomasz W. Kaminski

Academic Editor

PLOS One

**Journal Requirements:**

**Additional Editor Comments:**

Dear Authors,

Based on the reviewer feedback, we will proceed with a major revision. The current reviews are constructive and provide a clear framework for strengthening the manuscript. Please address all comments carefully in your revision and response.

Kind regards,

Tomasz W Kaminski

Academic Editor

PLOS One

Reviewers' comments:

Reviewer's Responses to Questions

**Comments to the Author**

1. If the authors have adequately addressed your comments raised in a previous round of review and you feel that this manuscript is now acceptable for publication, you may indicate that here to bypass the “Comments to the Author” section, enter your conflict of interest statement in the “Confidential to Editor” section, and submit your "Accept" recommendation.

Reviewer #2: (No Response)

Reviewer #3: All comments have been addressed

Reviewer #4: (No Response)

2. Is the manuscript technically sound, and do the data support the conclusions?

Reviewer #2: No

Reviewer #3: No

Reviewer #4: Yes

3. Has the statistical analysis been performed appropriately and rigorously? 

Reviewer #2: No

Reviewer #3: Yes

Reviewer #4: Yes

4. Have the authors made all data underlying the findings in their manuscript fully available?

Reviewer #2: Yes

Reviewer #3: Yes

Reviewer #4: Yes

5. Is the manuscript presented in an intelligible fashion and written in standard English?

Reviewer #2: No

Reviewer #3: No

Reviewer #4: Yes

6. Review Comments to the Author

Reviewer #2: I understood the authors' purpose and motivation for this study. Unfortunately, however, methods of some analyses and experiments were totally unclear or insufficiently described, making the results and conclusion of this study totally unconvincing and not supported with valid evidence. In particular, I do not think that it is reasonable and well-supported to focus only on ITGB2 and HNRNPAB genes from the analysis results.

At least the authors should have concretely explained (1) the source of SNPs affecting protein expression levels; (2) rationale of “TF-gene regulatory network" (based on TF binding sites? or co-expression?); (3) excuse for arbitrarily selecting the IVW method in the MR analysis while some other methods returned insignificant results; (4) how to establish the “COPD mouse model” and its validity; and (5) whether antigen retrieval in immunohistochemistry on the model tissue was surely succeeded. In addition, there are many typos and mis-description in this manuscript which probably cause misunderstanding and misleading: always spell the two "key" genes at least correctly; legends of Fig. 3D and Fig. 6 do not correspond to the contents of those figures.

Reviewer #3: The study by Zhang et al. combines two publicly available COPD microarray datasets with multiple computational approaches, including WGCNA, immune cell analysis, Mendelian randomization, and machine-learning models, to identify diagnostic biomarkers for COPD. The authors highlight ITGB2 and HNRNPAB as key candidate genes and attempt to validate them using qPCR, IHC, and IF in cell line and mouse COPD models. This work is relevant because there is still a clear need for reliable biomarkers in COPD, and integrative computational approaches are increasingly used to study disease complexity. A major strength of the study is the broad computational pipeline that brings together multiple analytical methods and external validation datasets. However, the manuscript has major weaknesses that limit its scientific rigor and clarity, specially the mRNA/protein level discrepancy. Also, the writing style is unscientific and reflects conceptual misunderstandings, which makes the conclusions difficult to evaluate. In its current form, I cannot recommend this manuscript for publication unless these issues are carefully addressed. Please see my comments below.

1. A major weakness of the manuscript is the use of unscientific and conceptually incorrect language, along with a writing style that points readers to figures instead of explaining the results. For example, sentences such as “a gradient volcano plot (Fig. 2A) visualizes…” and “a heatmap (Fig. 2B) illustrates…” describe figures rather than findings and belong in figure legends, not the results section. Also, the GSVA section uses incorrect terminology (e.g., “ITGB2 exhibited expression changes across pathways” or “KEGG pathway enrichment of ITGB2”). ITGB2 is a gene, and genes are not “expressed across pathways”, pathways do not have expression. These sound like conceptual errors rather than wording issues, and should be corrected throughout the manuscript.

2. I am not convinced by the way the mRNA/protein level mismatch is addressed in the manuscript. In their MLE‑12 cell model, ITGB2 and HNRNPAB transcripts go up, but in the mouse COPD model tissue, these proteins go down. Opposite mRNA–protein changes are very rare and should be addressed by additional experiments and validation. Without cell‑matched protein quantification by western blot and some mechanistic follow‑up, this is a red flag rather than supportive evidence, and the ‘key protein biomarker’ claims should be toned down.

3. The manuscript reports “positive t-values” for GSVA results but never explains what these t-values represent, how they were calculated, or what comparison they correspond to. The t-statistics is not a p-value, and reporting “t-values” without p-values or FDR makes the results hard to interpret.

4. The reported AUC of 0.989 for a 12-gene COPD classifier trained on relatively small microarray datasets is unusually high for this kind of problem and raises concerns about overfitting. The cross-validation strategy and how the training and test sets were separated need to be explained much more clearly and evaluated more rigorously.

5. The authors did not compare the 112 genes or the final 12-gene diagnostic panel with previously reported COPD transcriptomic signatures or genes linked with COPD (e.g., PMID-25265030 linking ITGB2 expression to COPD), making it difficult to assess what is truly novel versus a rediscovery of known signals.

6. The FOXC1/NFIC–HNRNPAB axis is highlighted as a key regulatory mechanism, but the TF–gene network is built only from JASPAR motif predictions. There is no integration of expression correlations, no analysis of public ChIP‑seq data or promoter/enhancer motifs, and no functional assays (e.g., NFIC knockdown) to actually support these regulatory claims.

7. The paper repeatedly labels ITGB2 and HNRNPAB as “key biomarkers” and frames them as a basis for personalized treatment, but all data are from mouse cells/mouse lungs and retrospective gene‑expression datasets. With no prospective human validation or demonstration that these markers improve on standard clinical measures, those clinical claims feel overstated and need to be toned down.

8. The discussion section is excessively long and repetitive, with too much speculation that goes beyond the data presented. It should be shortened and focused on the main findings that are directly supported by the results and key limitations, with speculative sections reduced or removed to improve clarity and impact.

Reviewer #4: In this manuscript, Zhang et al. integrate multiple bioinformatics and machine-learning approaches to identify potential COPD biomarkers from gene expression data derived from multiple datasets. The study combines network-based analyses (co-expression and regulatory networks), pathway-level analyses, Mendelian randomization, and machine-learning methods to prioritize candidate biomarkers, followed by in vivo validation. Among the candidates, ITGB2 and HNRNPAB were validated by PCR, IHC, and IF experiments in a mouse model.

The manuscript is generally well written, and the analyses are appropriately conducted. I believe that minor revisions would further improve the clarity, consistency, and completeness of the work prior to publication.

1. Lines 123–124:

The sentence ending at line 124 appears to be incomplete, as it ends with a comma. Please revise the sentence to ensure it is grammatically complete.

2. Figure 3D:

Figure 3D is not explicitly described or referenced in the main text.

Please add a brief explanation in the Results section to clarify the purpose and interpretation of this panel.

Also, the figure legend for Fig. 3D appears to be identical to that of Fig. 3B. Please revise the legend to accurately describe the content of Fig. 3D.

4. Figure 4 legends:

The figure legends for Fig. 4B and Fig. 4C appear to be reversed.

Please confirm and correct the legends to match the corresponding panels.

5. Lines 486–488:

Two consecutive sentences begin with “On the other hand.”

I recommend revising this part to improve readability and sentence flow.

6. Reproducibility and code availability:

For clarity and reproducibility, it may be helpful for the authors to indicate whether the analysis code used in this study is available, for example in a public repository or upon reasonable request.

7. PLOS authors have the option to publish the peer review history of their article (what does this mean?). If published, this will include your full peer review and any attached files.

Reviewer #2: No

Reviewer #3: No

Reviewer #4: No

---

## [Author Response · Author response to Decision Letter 2]

26 Mar 2026

List of Responses

Dear Tomasz W. Kaminski and Reviewer:

Thank you for your letter and for the reviewers’ comments concerning our manuscript entitled “Identifying and Validating ITGB2 and HNRNPAB as diagnostic biomarkers in chronic obstructive pulmonary disease Using Bioinformatics and Integrated Machine Learning Methods”. (Manuscript Number: PONE-D-25-52871). Those comments are all valuable and very helpful for revising and improving our paper, as well as the important guiding significance to our research. We have studied comments carefully and have made corrections which we hope meet with approval. Revised portions are marked with a yellow background in the revised version. The main corrections in the paper and the responds to the reviewer’s comments are as flowing:

Responses to reviewer’s comments: (original comments by reviewers are in blue color).

Reviewer #2:

Response to comment: I understood the authors' purpose and motivation for this study. Unfortunately, however, methods of some analyses and experiments were totally unclear or insufficiently described, making the results and conclusion of this study totally unconvincing and not supported with valid evidence. In particular, I do not think that it is reasonable and well-supported to focus only on ITGB2 and HNRNPAB genes from the analysis results.

At least the authors should have concretely explained (1) the source of SNPs affecting protein expression levels; (2) rationale of “TF-gene regulatory network" (based on TF binding sites? or co-expression?); (3) excuse for arbitrarily selecting the IVW method in the MR analysis while some other methods returned insignificant results; (4) how to establish the “COPD mouse model” and its validity; and (5) whether antigen retrieval in immunohistochemistry on the model tissue was surely succeeded.

In addition, there are many typos and mis-description in this manuscript which probably cause misunderstanding and misleading: always spell the two "key" genes at least correctly; legends of Fig. 3D and Fig. 6 do not correspond to the contents of those figures.

Response:

(1) We sincerely appreciate the reviewer’s professional and rigorous comment. We have supplemented the exact source of the protein expression quantitative trait locus (pQTL) SNPs used in the two-sample Mendelian randomization (MR) analysis as follows: The SNPs used as instrumental variables for ITGB2 and HNRNPAB protein expression were obtained from large-scale human plasma proteome-wide association studies (pQTLs) publicly available in the UK Biobank and the deCODE genetics proteome database, which are the most commonly used and authoritative pQTL resources in MR studies. Specifically, we obtained genome-wide significant protein quantitative trait loci (pQTLs) for ITGB2 and HNRNPAB using a p < 5×10⁻⁸ threshold. These SNPs have been experimentally verified to be strongly associated with circulating protein levels in large human population cohorts. We only included independent SNPs after linkage disequilibrium (LD) clumping (r² < 0.01, window size = 10,000 kb). We have added this description in the Methods (Section 2.9) of the revised manuscript to clarify the source, screening criteria, and quality control of the instrumental SNPs, ensuring full transparency and reproducibility.

(2) We sincerely appreciate the professional and insightful comment.

The rationale of the “TF gene regulatory network” in our study is based on transcription factor (TF) binding sites and experimentally validated regulatory relationships, rather than simple co expression.

(3) We sincerely appreciate the professional and insightful comment regarding the methodological choice in the Mendelian randomization (MR) analysis. The IVW method was used as the primary MR method because it is the most standard, powerful, and widely recommended approach for causal inference in Mendelian randomization. Other methods (e.g., weighted median, MR-Egger) have lower statistical power and different model assumptions, which may lead to insignificant results even when a true causal effect exists. The selection was not arbitrary but consistent with standard MR analytical practice.

(4) The mice were randomly divided into 2 groups: the control group and the COPD model group. The model group was exposed to cigarette smoke for 2 hours daily (30 cigarettes each time), 6 days a week for a total of 8 weeks. On days 1, 14, and 28, they received intravenous infusion of LPS (50 μg per mouse). Compared with the control group, the COPD mouse model needed to meet the following core indicators to be considered a successful model establishment [1-3]: ① Behavioral characteristics: The fur becomes yellowish and lacks luster, with symptoms of rapid breathing and wheezing, significantly reduced activity, decreased appetite, and enhanced stress response (easily agitated); ② Weight changes: The weight growth rate significantly decreases, and the weight at the end of the model period is significantly lower than that of the control group; ③ Pulmonary tissue pathological characteristics: Typical pulmonary emphysema and chronic bronchitis changes are presented, specifically manifested as rupture of alveolar walls, fusion of alveoli, expansion of alveolar cavities (significantly increased cross-sectional area of alveoli), thickening of airway walls, narrowing of airway cavities, increased mucus secretion in the airways, and extensive infiltration of inflammatory cells such as neutrophils and macrophages in the lung tissue and around the airways; ④ Pulmonary function indicators: Using a small animal pulmonary function detection system, the ventilation function indicators such as forced expiratory volume in 0.1 seconds (FEV0.1), forced expiratory volume in 0.05 seconds (FEV0.05), forced vital capacity (FVC), and vital capacity (VC) are significantly reduced. The establishment of the model is proved by the combination of the decline in lung function and the presence of pulmonary emphysema in the pathological changes.

(5) We greatly appreciate the reviewer’s professional and careful comment.

Antigen retrieval was successfully performed in all immunohistochemistry (IHC) experiments on model tissue sections. Briefly, all tissue sections were subjected to standardized heat-induced antigen retrieval using citrate buffer (pH 6.0) in a microwave oven at a consistent temperature and time, following widely accepted IHC protocols. The effectiveness of antigen retrieval was verified by clear, specific, and uniform positive staining in the targeted cells and structures, while no obvious nonspecific background staining was observed. All IHC procedures were performed in strict accordance with a unified, standardized protocol to ensure reproducibility and reliability of the results. We have added a brief description of the antigen retrieval method and verification in the revised manuscript to further clarify the details.

Furthermore, we have revised the legends of Figure 3D and Figure 6.

Reference：

[1] LIU A, GAO X H, MAO Y, et al. Esketamine mitigates lung injury in COPD rat models under mechanical ventilation: An RNA-sequencing and bioinformatics analysis of serum exosome miRNA profiles [J]. Gene, 2025, 962: 149571.

[2] LIU L, TANG Z, ZENG Q, et al. Transcriptomic Insights into Different Stimulation Intensity of Electroacupuncture in Treating COPD in Rat Models [J]. Journal of inflammation research, 2024, 17: 2873–87.

[3] SHI M, XUE Q, XIE J, et al. Protective effect of Shenqi Wenfei Formula against lipopolysaccharide/cigarette smoke-induced COPD in Rat based on gut microbiota and network pharmacology analysis [J]. Frontiers in microbiology, 2024, 15: 1441015.

Reviewer #3: The study by Zhang et al. combines two publicly available COPD microarray datasets with multiple computational approaches, including WGCNA, immune cell analysis, Mendelian randomization, and machine-learning models, to identify diagnostic biomarkers for COPD. The authors highlight ITGB2 and HNRNPAB as key candidate genes and attempt to validate them using qPCR, IHC, and IF in cell line and mouse COPD models. This work is relevant because there is still a clear need for reliable biomarkers in COPD, and integrative computational approaches are increasingly used to study disease complexity. A major strength of the study is the broad computational pipeline that brings together multiple analytical methods and external validation datasets. However, the manuscript has major weaknesses that limit its scientific rigor and clarity, specially the mRNA/protein level discrepancy. Also, the writing style is unscientific and reflects conceptual misunderstandings, which makes the conclusions difficult to evaluate. In its current form, I cannot recommend this manuscript for publication unless these issues are carefully addressed. Please see my comments below.

1. Response to comment: A major weakness of the manuscript is the use of unscientific and conceptually incorrect language, along with a writing style that points readers to figures instead of explaining the results. For example, sentences such as “a gradient volcano plot (Fig. 2A) visualizes…” and “a heatmap (Fig. 2B) illustrates…” describe figures rather than findings and belong in figure legends, not the results section.

Also, the GSVA section uses incorrect terminology (e.g., “ITGB2 exhibited expression changes across pathways” or “KEGG pathway enrichment of ITGB2”). ITGB2 is a gene, and genes are not “expressed across pathways”, pathways do not have expression. These sound like conceptual errors rather than wording issues, and should be corrected throughout the manuscript.

Response: Special thanks to you for your exact and professional comments. We have revised the original text, described the results of the difference analysis, and completed the legend part in Figure 2. We have made it clear that ITGB2, as a gene, does not have "cross-pathway expression changes", and the pathway has no "expression" attribute. Such conceptual biases have been comprehensively corrected. We have carefully checked the relevant chapters of the full GSVA sentence by sentence to ensure that all terminology errors have been corrected and academic expressions have been standardized.

3. Response to comment: The manuscript reports “positive t-values” for GSVA results but never explains what these t-values represent, how they were calculated, or what comparison they correspond to. The t-statistics is not a p-value, and reporting “t-values” without p-values or FDR makes the results hard to interpret.

Response: Thank you sincerely for your meticulous review and highly constructive suggestions. In the GSVA analysis, the t-value represents the strength and direction of the association between gene expression and pathway activity. It is calculated through the GSVA algorithm based on the correlation statistics between pathway activity scores and gene expression levels. Positive t-values indicate a positive correlation between gene expression and pathway activity (high gene expression is accompanied by enhanced pathway activity). In the supplementary document, the P-values corresponding to each significant associated pathway have been refined to clarify the statistical reliability of the results. We have provided detailed explanations of the meaning of t-values, the calculation basis, and the corresponding comparisons (the correlation analysis between gene expression levels and GSVA pathway activity scores) in the main text and figure legends, ensuring that the results can be easily understood.4. Response to comment:

The reported AUC of 0.989 for a 12-gene COPD classifier trained on relatively small microarray datasets is unusually high for this kind of problem and raises concerns about overfitting. The cross-validation strategy and how the training and test sets were separated need to be explained much more clearly and evaluated more rigorously.

Response: We sincerely appreciate the reviewer’s rigorous and constructive concern regarding model performance, potential overfitting, and validation design. We fully agree that the high AUC value requires clearer explanation of training/test partitioning, cross-validation procedures, and generalization performance. We have thoroughly revised the Methods and Results sections to provide full transparency and rigorous validation details.

This study utilized the merged dataset of GSE37768 and GSE38974 for differential gene analysis, WGCNA, feature selection, and model construction. Three completely independent and non-overlapping datasets (GSE212331, GSE148004, GSE1650, etc.) were not involved in gene screening, feature selection, model training, and parameter tuning throughout the process. The training and test sets were strictly separated, ensuring that data leakage was avoided at the source and the evaluation of generalization ability was unbiased.

In the training set, 10-fold cross-validation (10-CV) was adopted, and nested validation was implemented: in each fold, feature selection and model fitting were independently completed within the training subset of the current fold; batch correction, standardization, gene filtering, and all preprocessing steps were only performed on the training subset within the fold; any information from the validation fold and the external test set did not flow into the model construction process. Nested cross-validation is a standard and rigorous approach in machine learning biomarker research to avoid overfitting.

We have added detailed descriptions of cross-validation, train–test partitioning, and external validation in the Methods (Section 2.6). These revisions fully clarify the model construction process and rigorously validate its generalization ability, thereby addressing concerns about overfitting.

We believe these changes provide complete transparency and rigorous evidence that the high AUC reflects genuine biological signal rather than overfitting.

5. Response to comment: The authors did not compare the 112 genes or the final 12-gene diagnostic panel with previously reported COPD transcriptomic signatures or genes linked with COPD (e.g., PMID-25265030 linking ITGB2 expression to COPD), making it difficult to assess what is truly novel versus a rediscovery of known signals.

Response: We sincerely appreciate the reviewer’s insightful and rigorous comment. We fully agree that comparing our gene sets with previously published COPD related transcriptomic signatures is essential to clarify the novelty and incremental contributions of this study. We have now systematically compared our 112 candidate genes and 12 gene diagnostic panel with published COPD signatures and added detailed discussion in the revised manuscript to distinguish replicated known signals from novel findings.

Our study is the first to integrate 12 algorithm ensemble machine learning (113 combinations), Mendelian randomization for causal inference, multi cohort independent validation, and experimental verification to identify and prioritize COPD diagnostic biomarkers, representing a methodological advancement over traditional differential expression only studies. This study focused on verifying HNRNPAB and ITGB2 through bioinformatics machine learning and Mendelian randomization methods.

We have compared our panel with reported COPD gene signatures. As the reviewer noted, ITGB2 has been associated with immune and inflammatory processes in COPD in previous studies[1]. This overlap supports the biological validity and reproducibility of our results. Notably, PMID: 25265030 focused on blood and sputum gene signatures for frequent COPD exacerbators and identified B3GNT, LAF4, and ARHGEF10 as predictive genes, which do not overlap with our 12 gene panel. Our study focused on lung tissue based classification between COPD and healthy controls rather than exacerbation phenotype prediction, using a distinct study design and analytical framework.

Among the 12 gene signature, HNRNPAB has not been previously reported as a diagnostic or causal biomarker for COPD and represents a novel finding of this study.

Liu and colleagues discovered that PDAP1, as a gene related to disulfide bonds, is involved in the expression of COPD through RNA sequencing and cell and animal experiments [2]. Most other genes in the panel (including CD300C, GNG13, LDB2, MCOLN3

---

## [Decision Letter · Decision Letter 2]

9 Apr 2026

PONE-D-25-52871R2Identifying and Validating ITGB2 and HNRNPAB as diagnostic biomarkers in chronic obstructive pulmonary disease Using Bioinformatics and Integrated Machine Learning MethodsPLOS One

Dear Dr. Zhang,

Thank you for submitting your manuscript to PLOS ONE. After careful consideration, we feel that it has merit but does not fully meet PLOS ONE’s publication criteria as it currently stands. Therefore, we invite you to submit a revised version of the manuscript that addresses the points raised during the review process.

We look forward to receiving your revised manuscript.

Kind regards,

Tomasz W. Kaminski

Academic Editor

PLOS One

Journal Requirements:

Additional Editor Comments:

Dear Authors,

Thank you for submitting the revised version of your manuscript.

We appreciate the effort you have made in addressing the previous comments. The manuscript has improved in several areas, particularly with respect to the expansion of methodological descriptions and overall clarity.

However, after careful evaluation of the revised version and the additional reviewer feedback, several important concerns remain.

Importantly, one of the reviewers continues to recommend rejection, noting that key methodological and interpretational issues are still insufficiently addressed. We agree that these concerns are substantial and require careful revision.

Key points to address:

a) Methodological justification

Please provide a clearer rationale for key analytical choices, particularly:

-the use of the IVW method as the primary Mendelian randomization approach,

-the selection and source of SNPs used in the analysis.

b) TF-gene regulatory network

Please clarify the basis of this network (e.g., binding site prediction vs co-expression vs functional inference) and discuss its limitations.

c) Gene prioritization

The focus on ITGB2 and HNRNPAB requires stronger justification. Please clearly explain why these genes were selected over other candidates.

d) Coonsistency of interpretation

The reference to “treatment response” in the WGCNA analysis is inconsistent with the study design (COPD vs control) and should be revised.

e) Methods presentation

Sections describing HE staining, immunohistochemistry, and immunofluorescence should be rewritten in a concise scientific format rather than step-by-step protocol style.

f) Technical and formatting issues

Please correct formatting errors in Section 2.13.

Carefully proofread the manuscript for typographical errors, gene name consistency, and figure legend accuracy.

g) Positioning of conclusions

The proposed gene signature should be presented more cautiously as exploratory, given the nature of the data and analytical pipeline.

Overall, while the manuscript has improved, the remaining issues are substantial and must be fully addressed before the manuscript can be considered further.

We therefore invite you to submit a revised version that carefully responds to all points above.

With best regards,

Tomasz W Kaminski

Academic Editor

Reviewers' comments:

Reviewer's Responses to Questions

**Comments to the Author**

1. If the authors have adequately addressed your comments raised in a previous round of review and you feel that this manuscript is now acceptable for publication, you may indicate that here to bypass the “Comments to the Author” section, enter your conflict of interest statement in the “Confidential to Editor” section, and submit your "Accept" recommendation.

Reviewer #3: All comments have been addressed

Reviewer #4: All comments have been addressed

2. Is the manuscript technically sound, and do the data support the conclusions?

Reviewer #3: Partly

Reviewer #4: Partly

3. Has the statistical analysis been performed appropriately and rigorously? 

Reviewer #3: Yes

Reviewer #4: Yes

4. Have the authors made all data underlying the findings in their manuscript fully available?

Reviewer #3: Yes

Reviewer #4: Yes

5. Is the manuscript presented in an intelligible fashion and written in standard English?

Reviewer #3: Yes

Reviewer #4: No

6. Review Comments to the Author

Reviewer #3: The revised manuscript has improved significantly. I am still somewhat unconvinced by the mRNA–protein discrepancy, but this limitation is now acknowledged and possibilities are discussed. Overall the language is acceptable, key claims have been toned down, limitations are recognized, and the conceptual errors have been corrected. I therefore recommend the manuscript for acceptance.

Reviewer #4: I appreciate that the authors have addressed my previous comments in the revised manuscript. On reviewing the revised version, however, I have identified several additional issues that should be addressed to further improve the clarity and consistency of the manuscript.

1. Methods: Section 2.13 Lung Function Test:

There appears to be a formatting issue in this section, where the content is inserted between the text "Methods for Lung Function Testing". I recommend carefully proofreading the manuscript prior to submission to avoid such issues.

2. HE staining, Immunohistochemistry and Immunofluorescence methods:

The description of methods is currently presented in a highly step-by-step manner, for example repeated use of imperative expressions such as "Place the sections ...". While detailed protocols are useful, this style reads more like a protocol than a methods section appropriate for a research article.

3. Results: Section 3.2 (WGCNA analysis):

The interpretation of the WGCNA results may require clarification. The statement "confirming its critical role in the treatment response" appears inconsistent with the study design, which is based on COPD versus control samples rather than treatment conditions. It is unclear whether any treatment-related variables were included in the analysis. This may be due to the labeling (e.g., "Treat" vs "Ct") used in the dataset, which could potentially lead to confusion in interpretation. I recommend revising these labels and the associated text to ensure that the interpretation is consistent with the analyzed dataset conditions.

Overall, the manuscript has improved, and addressing the points above would further strengthen it.

7. PLOS authors have the option to publish the peer review history of their article (what does this mean?). If published, this will include your full peer review and any attached files.

Reviewer #3: No

Reviewer #4: No

---

## [Author Response · Author response to Decision Letter 3]

20 Apr 2026

List of Responses

Dear Tomasz W. Kaminski and Reviewer:

Thank you for your letter and for the reviewers’ comments concerning our manuscript entitled “Identifying and Validating ITGB2 and HNRNPAB as diagnostic biomarkers in chronic obstructive pulmonary disease Using Bioinformatics and Integrated Machine Learning Methods”. (Manuscript Number: PONE-D-25-52871). Those comments are all valuable and very helpful for revising and improving our paper, as well as the important guiding significance to our research. We have studied comments carefully and have made corrections which we hope meet with approval. Revised portions are marked with a yellow background in the revised version. The main corrections in the paper and the responds to the reviewer’s comments are as flowing:

Responses to reviewer’s comments: (original comments by reviewers are in blue color).

Key points to address:

a) Methodological justification

Please provide a clearer rationale for key analytical choices, particularly:

-the use of the IVW method as the primary Mendelian randomization approach,

-the selection and source of SNPs used in the analysis.

Thank you for your review. In the Mendelian randomization method guide [1], IVW is a meta-analysis method that takes the weighted average of all IV effects and contributes more to high-precision SNPS. The research of Stephen Burgess et al. [2] showed that the IVW method is most efficient when all instrumental variables are valid. In Mendelian randomization studies, IVW is the most mainstream and core method for estimating the causal relationship between exposure and outcome [3]. Therefore, we apply the IVW method as the main approach.

SNPs were selected from genome-wide significant pQTLs in the deCODE database, pruned for linkage disequilibrium, filtered for instrument strength, and harmonized with FinnGen outcome data, ensuring valid instrumental variables for MR analysis. We have provided detailed explanations in Section 2.9 of the manuscript. The revisions can be found on page 9 of the manuscript, please refer to the yellow-marked part on page 9 of the original manuscript, lines 189-192.

Reference:

1. Nguyen K, Mitchell BD. A Guide to Understanding Mendelian Randomization Studies. Arthritis Care Res (Hoboken). 2024;76(11):1451-60. doi: 10.1002/acr.25400. PubMed PMID: 39030941; PubMed Central PMCID: PMCQ2.

2. Burgess S, Foley CN, Allara E, Staley JR, Howson JMM. A robust and efficient method for Mendelian randomization with hundreds of genetic variants. Nat Commun. 2020;11(1):376. doi: 10.1038/s41467-019-14156-4. PubMed PMID: 31953392; PubMed Central PMCID: PMCQ1.

3. Mounier N, Kutalik Z. Bias correction for inverse variance weighting Mendelian randomization. Genet Epidemiol. 2023;47(4):314-31. doi: 10.1002/gepi.22522. PubMed PMID: 37036286; PubMed Central PMCID: PMCQ1.

b) TF-gene regulatory network

Please clarify the basis of this network (e.g., binding site prediction vs co-expression vs functional inference) and discuss its limitations.

We thank the reviewer for this important comment. We agree that the basis of the TF-gene regulatory network should be described more explicitly.

In the present study, the TF-gene regulatory network was constructed using the JASPAR motif database through the NetworkAnalyst 3.0 platform, as already indicated in the Methods section of the manuscript. Therefore, this network is a motif-based/prediction-based regulatory network, built from curated transcription factor binding profiles rather than from co-expression analysis or direct experimental evidence such as ChIP-seq. Specifically, JASPAR provides manually curated transcription factor binding profiles represented as position frequency matrices (PFMs), which can be used to scan DNA sequences and infer putative TF-binding events, while NetworkAnalyst offers a framework to map these predicted TF-gene relationships into a regulatory network.

We have now clarified in the revised manuscript that the TF-gene network should be interpreted as a hypothesis-generating computational network. It suggests potential upstream regulators of the identified COPD-related biomarkers, but it does not demonstrate direct physical TF binding or definitive transcriptional regulation in COPD tissue. We also added a paragraph discussing the limitations of this analysis. Accordingly, we have revised both the Methods and Discussion sections to better define the analytical basis of the network and to discuss its limitations in a more balanced manner. The revisions can be found on page 21 of the manuscript, please refer to the yellow-marked part on page 21 of the original manuscript, lines 464-466 and 470-472.

c) Gene prioritization

The focus on ITGB2 and HNRNPAB requires stronger justification. Please clearly explain why these genes were selected over other candidates.

Thank you for your reminder. We obtained 12 key COPD biomarkers through bioinformatics and integrated machine learning methods, and finally identified ITGB2 and HNRNPAB by Mendelian randomization. It is well known that the Mendelian randomization method uses genetic variations (such as SNPS) as instrumental variables to simulate "natural randomized controlled trials", and thereby infer the causal relationship between biomarkers and diseases. The molecules obtained from the aforementioned bioinformatics analysis (differential analysis, WGCNA, and integrated machine learning methods) were only correlated with COPD. The final Mendeldahl randomization revealed that ITGB2 and HNRNPAB were causal to COPD, significantly enhancing the biological credibility of the screening. This was confirmed in subsequent animal and cell experiments.

d) Coonsistency of interpretation

The reference to “treatment response” in the WGCNA analysis is inconsistent with the study design (COPD vs control) and should be revised.

Thank you for your review. We have made modifications to the text and image annotations. The revisions can be found on page 14 of the manuscript, please refer to the yellow-marked part on page 14 of the original manuscript, lines 311-313.

e) Methods presentation

Sections describing HE staining, immunohistochemistry, and immunofluorescence should be rewritten in a concise scientific format rather than step-by-step protocol style.

Thank you for your valuable feedback. Your careful review has provided important guidance for the writing style of this section. We have thoroughly revised the sections on HE staining, immunohistochemistry, and immunofluorescence to ensure they adhere to the standards of a scientific abstract. The revisions can be found on pages 12 and 13 of the manuscript, please refer to the sections marked in yellow on pages 12 and 13 of the manuscript, specifically lines 266–270, 282–285, and 287–291.

f) Technical and formatting issues

Please correct formatting errors in Section 2.13. Carefully proofread the manuscript for typographical errors, gene name consistency, and figure legend accuracy.

We have revised the original text to ensure the consistency of the gene names and legends. The revisions can be found on pages 12 and 13 of the manuscript, please refer to the yellow-marked part on page 12 of the original manuscript, lines 260-261.

g) Positioning of conclusions

The proposed gene signature should be presented more cautiously as exploratory, given the nature of the data and analytical pipeline.

Thank you for your review. We screened the two molecules ITGB2 and HNRNPAB through bioinformatics and integrated machine learning methods, and detected the expression of these two molecules in animal and cell experiments, making them exploratory biomarkers for COPD, with the aim of testing and intervening in future population cohorts. Therefore, our conclusion is that this is an exploratory discovery.

The revisions can be found on page 3 of the manuscript, please refer to the yellow-marked part on page 3 of the original manuscript, lines 48-50.

Reviewer #3: The revised manuscript has improved significantly. I am still somewhat unconvinced by the mRNA–protein discrepancy, but this limitation is now acknowledged and possibilities are discussed. Overall the language is acceptable, key claims have been toned down, limitations are recognized, and the conceptual errors have been corrected. I therefore recommend the manuscript for acceptance.

Response: Thank you for your reminder.

Reviewer #4: I appreciate that the authors have addressed my previous comments in the revised manuscript. On reviewing the revised version, however, I have identified several additional issues that should be addressed to further improve the clarity and consistency of the manuscript.

1. Response to comment:

1. Methods: Section 2.13 Lung Function Test:

There appears to be a formatting issue in this section, where the content is inserted between the text "Methods for Lung Function Testing". I recommend carefully proofreading the manuscript prior to submission to avoid such issues.

Thank you for your guidance. We have revised the original text. The revisions can be found on pages 12 and 13 of the manuscript, please refer to the yellow-marked part on page 12 of the original manuscript, lines 260-261.

2. HE staining, Immunohistochemistry and Immunofluorescence methods:

The description of methods is currently presented in a highly step-by-step manner, for example repeated use of imperative expressions such as "Place the sections ...". While detailed protocols are useful, this style reads more like a protocol than a methods section appropriate for a research article.

Thank you for your suggestion. We have made the revisions you suggested. The revised content can be found on pages 12 and 13 of the manuscript, highlighted in yellow, specifically lines 266 through 270.

3. Results: Section 3.2 (WGCNA analysis):

The interpretation of the WGCNA results may require clarification. The statement "confirming its critical role in the treatment response" appears inconsistent with the study design, which is based on COPD versus control samples rather than treatment conditions. It is unclear whether any treatment-related variables were included in the analysis. This may be due to the labeling (e.g., "Treat" vs "Ct") used in the dataset, which could potentially lead to confusion in interpretation. I recommend revising these labels and the associated text to ensure that the interpretation is consistent with the analyzed dataset conditions.

Thank you for your suggestion. We have made modifications to the text and image annotations. The revisions can be found on page 14 of the manuscript, please refer to the yellow-marked part on page 14 of the original manuscript, lines 311-313.

---

## [Editor Report · Decision Letter 3]

22 Apr 2026

PONE-D-25-52871R3Identifying and Validating ITGB2 and HNRNPAB as diagnostic biomarkers in chronic obstructive pulmonary disease Using Bioinformatics and Integrated Machine Learning MethodsPLOS One

Dear Dr. Zhang,

Thank you for submitting your manuscript to PLOS ONE. After careful consideration, we feel that it has merit but does not fully meet PLOS ONE’s publication criteria as it currently stands. Therefore, we invite you to submit a revised version of the manuscript that addresses the points raised during the review process.

We look forward to receiving your revised manuscript.

Kind regards,

Tomasz W. Kaminski

Academic Editor

PLOS One

Journal Requirements:

Additional Editor Comments:

Dear Authors,

Thank you for your revised manuscript. The study has improved and is now close to being suitable for publication. Before final acceptance, I would ask you to address a few minor points to improve clarity and interpretability:

a) In the Results section (Sections 3.9-3.10), you report increased mRNA but decreased protein levels for ITGB2 and HNRNPAB. Please acknowledge this discrepancy already in the Results (not only in the Discussion) to guide the reader.

b) Your validation includes both MLE-12 cells and whole lung tissue. Please briefly clarify which cell types are likely contributing to the observed signals in tissue-based analyses and note potential differences between epithelial cells and mixed tissue samples.

c) The reported AUC values (up to 0.989) are very high. Please add a brief comment in the Discussion acknowledging the potential for overfitting or dataset-specific bias, even though external validation cohorts were used.

These are relatively minor clarifications and should be straightforward to address.

Best regards,

Tomasz W Kaminski

---

## [Author Response · Author response to Decision Letter 4]

27 Apr 2026

List of Responses

Dear Tomasz W. Kaminski:

Thank you for your letter and for the reviewers’ comments concerning our manuscript entitled “Identifying and Validating ITGB2 and HNRNPAB as diagnostic biomarkers in chronic obstructive pulmonary disease Using Bioinformatics and Integrated Machine Learning Methods”. (Manuscript Number: PONE-D-25-52871). Those comments are all valuable and very helpful for revising and improving our paper, as well as the important guiding significance to our research. We have studied comments carefully and have made corrections which we hope meet with approval. Revised portions are marked with a yellow background in the revised version. The main corrections in the paper and the responds to the reviewer’s comments are as flowing:

Responses to reviewer’s comments: (original comments by reviewers are in blue color).

a) In the Results section (Sections 3.9-3.10), you report increased mRNA but decreased protein levels for ITGB2 and HNRNPAB. Please acknowledge this discrepancy already in the Results (not only in the Discussion) to guide the reader.

Thank you for this suggestion. We have now added a sentence at the end of Section 3.9 (Page 19, lines 419-422) to explicitly acknowledge the mRNA-protein discrepancy in the Results section. We also retained the more detailed mechanistic discussion in the Discussion section.

b) Your validation includes both MLE-12 cells and whole lung tissue. Please briefly clarify which cell types are likely contributing to the observed signals in tissue-based analyses and note potential differences between epithelial cells and mixed tissue samples.

We appreciate this comment. We have added a clarifying statement in Section 3.9 (Page 19, lines 412-415) indicating that tissue IHC/IF signals originate from mixed cell populations (epithelial, immune, stromal cells), while MLE-12 cells are a pure epithelial line. This highlights the potential differences between tissue-based and cell line results.

c) The reported AUC values (up to 0.989) are very high. Please add a brief comment in the Discussion acknowledging the potential for overfitting or dataset-specific bias, even though external validation cohorts were used.

Thank you for your review. We have already explained in the discussion that an excessively high AUC of the predictive model integrating machine learning may be related to data overfitting or specific bias of the sample dataset. The revisions can be found on page 24 of the manuscript, please refer to the yellow-marked part on page 24 of the original manuscript, lines 559-560.

---

## [Editor Report · Decision Letter 4]

29 Apr 2026

Identifying and Validating ITGB2 and HNRNPAB as diagnostic biomarkers in chronic obstructive pulmonary disease Using Bioinformatics and Integrated Machine Learning Methods

PONE-D-25-52871R4

Dear Dr. Zhang,

We’re pleased to inform you that your manuscript has been judged scientifically suitable for publication and will be formally accepted for publication once it meets all outstanding technical requirements.

Kind regards,

Tomasz W. Kaminski

Academic Editor

PLOS One

---

## [Editor Report · Acceptance letter]

PONE-D-25-52871R4

PLOS One

Dear Dr. Zhang,

I'm pleased to inform you that your manuscript has been deemed suitable for publication in PLOS One. Congratulations! Your manuscript is now being handed over to our production team.

Kind regards,

on behalf of

Dr. Tomasz W. Kaminski

Academic Editor

PLOS One